# ARFlow: Auto-regressive Optical Flow Estimation for Arbitrary-Length Videos via Progressive Next-Frame Forecasting

**Jiuming Liu**[1][*], **Mengmeng Liu**[2][*], **Siting Zhu**[1], **Yunpeng Zhang**[3], **Jiangtao Li**[3],
**Michael Ying Yang**[4], **Francesco Nex**[2], **Hao Cheng**[2][†], **Hesheng Wang**[1][†]
[1]Shanghai Jiao Tong University   [2]University of Twente   [3]Phigent Robotics   [4]University of Bath
`liujiuming123@gmail.com, m.liu-1@utwente.nl`

## Abstract

Optical flow estimation is a fundamental computer vision task that predicts per-pixel displacements from consecutive images. Recent works attempt to exploit temporal cues to improve the estimation performance. However, their temporal modeling is restricted to short video sequences due to the unaffordable computational burden, thereby suffering from restricted temporal receptive fields. Moreover, their group-wise paradigm in one forward pass undermines inter-group information exchange, leading to modest performance improvement. To address these problems, we propose a novel multi-frame optical flow network based on an auto-regressive paradigm, named ARFlow. Unlike previous multi-frame methods, our method can be scalable to arbitrary-length videos with marginal computational overhead. Specifically, we design an Auto-regressive Flow Initialization (AFI) module and an Auto-regressive Multi-stride Flow Refinement (AMFR) module to forecast the next-frame flow based on multi-stride history observations. Our ARFlow achieves state-of-the-art performance, ranking 1st on both KITTI-2015 and Spring official benchmarks and 2nd on the MPI-Sintel (Final) benchmark among all open-sourced methods. Furthermore, due to the auto-regressive nature, our method can generalize to arbitrary video length with a constant GPU memory usage of 2.1GB.

## 1 Introduction

Optical flow indicates the 2D displacement field of each pixel predicted from consecutive video frames, playing a key role in various downstream applications such as video inpainting (Xu et al., 2019; Li et al., 2022), dynamic scene reconstruction (LU et al., 2025; Zhu et al., 2024; Deng et al., 2025b; Chan et al., 2026; Li et al., 2023), and video generation (Liang et al., 2024a;b).

With the development of advanced model architectures (Deng et al., 2025c), optical flow estimation has witnessed remarkable success, evolving from CNN-based backbones (Sun et al., 2018; Teed & Deng, 2020) to transformer-based ones (Huang et al., 2022; Xu et al., 2022); from discriminative methods (Zhao et al., 2024) to generative ones (Luo et al., 2024; Saxena et al., 2023; Liu et al., 2024a; 2025c). However, these methods typically formulate the task with a pairwise paradigm, which neglects intrinsic temporal coherence within video sequences, leading to suboptimal performance and poor reasoning ability in occluded areas (Chen et al., 2023; Dong & Fu, 2024; Liu et al., 2023c).

Recently, there has been an increasing research focus on multi-frame optical flow estimation from video sequences (Shi et al., 2023a; Sun et al., 2024; Bargatin et al., 2025; Dong & Fu, 2024), as shown in Fig. 1. However, we observe that concurrent multi-frame methods commonly fail to exploit sufficient temporal cues because of the following: (1) *Limited temporal receptive fields:* Existing multi-frame settings typically segment the entire video into multiple groups with a fixed size, e.g., three frames in MemFlow (Dong & Fu, 2024) (Fig. 1 (A)), four frames in StreamFlow (Sun et al., 2024) (Fig. 1 (B)). This group-wise segmentation fails to capture long-range dependencies

---

[*]These authors contributed equally.   [†]Corresponding authors: Hao Cheng, Hesheng Wang.

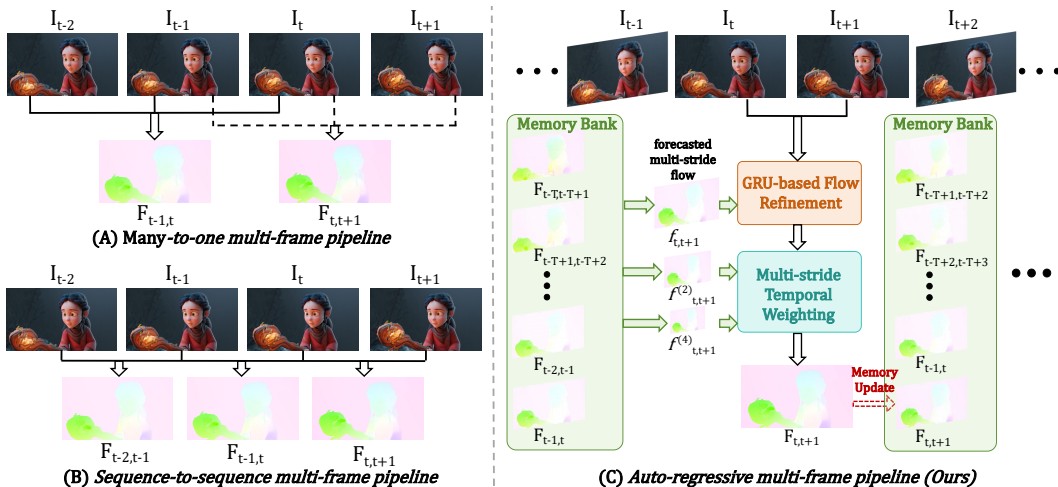

Figure 1: Comparison with previous multi-frame optical flow methods.

throughout the total video length. (2) *Heavy computational overhead and memory usage:* Vide-oFlow (Shi et al., 2023a) inputs overlapping clips to estimate multiple flows. MemFlow (Dong & Fu, 2024) utilizes many image inputs to strengthen the current estimation. Both approaches repeat feature extraction and context encoder several times in different groups, impeding real-time applications. StreamFlow (Sun et al., 2024) suffers from high GPU memory usage, as shown in Fig. 2. This inefficiency further leads to poor scalability, allowing only a limited number of frame inputs. (3) *Lack of multi-stride temporal modeling:* Prior methods typically focus on single-stride temporal modeling with short-term motions. However, both long-range and short-term motions are crucial for optical flow: long-range motions facilitate recognizing occlusions and out-of-boundary pixels across frames (Shi et al., 2023a), while short-term motions capture subtle variations in adjacent frames.

To address these problems, we propose a novel multi-frame optical flow estimation pipeline (ARFlow) using an auto-regressive prediction paradigm in Fig. 1 (C). Specifically, an Auto-regressive Flow Initialization module (AFI) is developed to retrieve multiple history flow estimates from the memory bank and forecast the next-frame initial flow based on the history ones. This process will introduce an accurately initialized flow, which significantly improves the performance and also decreases the iteration numbers in subsequent refinement modules, enabling higher efficiency. After obtaining the initial flow, another Auto-regressive Multi-

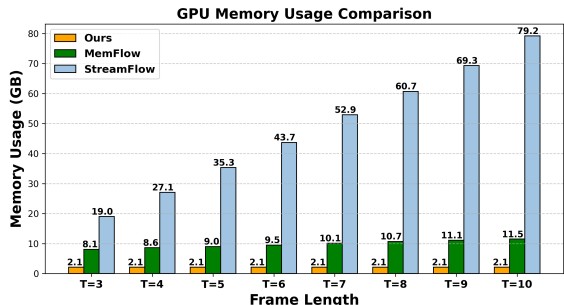

Figure 2: Previous group-wise multi-frame methods suffer from a significant memory usage increase with longer frame length. In contrast, our method maintains nearly constant GPU consumption.

stride Flow Refinement module (AMFR) is also designed by incorporating multi-granularity temporal information from diverse intervals to combine both long-term and short-term motions. Unlike previous multi-frame works that segment videos into multiple groups, we utilize the entire video sequence as the network input, where sequential images are delivered frame by frame. When a new frame is input, its features are extracted and correlated with previous history flows to predict the current flow. This paradigm breaks the limitation of local receptive fields, possessing strong scalability to arbitrary frame lengths with no memory usage increase in Fig. 2. **Please refer to the supplementary videos for scalable optical flow estimation on arbitrary-length inputs.**

Overall, the main contributions of our method are as follows:

- We design a novel multi-frame optical flow estimation method, named ARFlow, based on the auto-regressive paradigm. Through progressively forecasting the next-frame flow, our method can be scalable to arbitrary-length videos.

- An Auto-regressive Flow Initialization module (AFI) is developed to retrieve multiple history flows from memory banks and forecast the next-frame initial flow based on these history estimates.

- An Auto-regressive Multi-stride Flow Refinement module (AMFR) is also designed to further refine the initial flow by incorporating both long-term and short-term temporal cues.

- Extensive experiments on MPI-Sintel (Butler et al., 2012), Spring (Mehl et al., 2023b), and KITTI-2015 (Geiger et al., 2013) datasets demonstrate the state-of-the-art performance of our proposed ARFlow. Moreover, due to the accurately predicted initial flows, the number of iterations in the following refinement module is decreased, enabling lower memory usage compared with prior multi-frame methods.

## 2 RELATED WORK

**Pairwise Optical Flow Estimation.** Traditional methods (Black & Anandan, 1993; Bruhn et al., 2005; Horn & Schunck, 1981) typically formulate optical flow estimation as an optimization problem which maximizes the visual similarity between two frames. FlowNet (Dosovitskiy et al., 2015) regresses the optical flow with Convolutional Neural Networks (CNNs) in an end-to-end manner. PWC-Net (Sun et al., 2018) proposes a Pyramid, Warping, Cost volume method for coarse-to-fine flow estimations. To address the problem of small fast-moving objects, RAFT (Teed & Deng, 2020) designs a recurrent iteration module to improve estimation accuracy. Subsequent works (Ranjan & Black, 2017; Hui et al., 2018; Yang & Ramanan, 2019; Hui et al., 2020; Jiang et al., 2021; Zhang et al., 2021; Sun et al., 2022; Jahedi et al., 2024; Wang et al., 2024b) progressively refine model architectures in a coarse-to-fine or iterative manner. Various transformer-based methods (Huang et al., 2022; Shi et al., 2023b; Xu et al., 2022; Liu et al., 2023b; Shan et al., 2021; Sui et al., 2022; Luo et al., 2023) are then developed to enlarge the matching receptive field. Some methods Liu et al. (2022; 2023a) also incorporate scene flow estimation Jiang et al. (2024); Liu et al. (2025d); Zhang et al. (2024); Liu et al. (2025b); Zhang et al. (2025); Liu et al. (2024c;b); Feng et al. (2024) by joint optimization. Nevertheless, these two-frame optical flow methods fail to take intrinsic temporal cues across frames into consideration.

**Multi-Frame Optical Flow Estimation.** The research for multi-frame flow estimation has gained remarkable advances in recent years. RAFT (Teed & Deng, 2020) utilizes a "warm-start" strategy, which initializes the current flow by warping previous flow estimates. Some self-supervised methods (Liu et al., 2019; Hur & Roth, 2021) are also proposed to retrieve temporal information by CNN or LSTM. VideoFlow (Shi et al., 2023a) calculates the bi-directional flows with a three-frame setting and designs a motion propagation module to exchange information across different triplets. Splat-Flow (Wang et al., 2024a) introduces a differential splatting transformation to align motion features from previous timestamps. StreamFlow (Sun et al., 2024) proposes an in-batch estimation strategy, simultaneously estimating all successive flows. More recently, memory mechanisms (Dong & Fu, 2024; Bargatin et al., 2025) have been introduced to store previous flow-related features and further improve the current estimation. However, these prior multi-frame methods can only process segmented groups with limited sizes (3-5 frames), impeding sufficient temporal modeling. Also, they suffer from heavy computational burdens and memory usage with increasing video lengths. In this paper, our ARFlow resorts to the auto-regressive transformer by progressively forecasting the next frame's motion, which is scalable to arbitrary video lengths while keeping low memory usage.

**Auto-regressive Video Generation.** Recently, auto-regressive models benefiting from scaling laws (Henighan et al., 2020) have gained great success in the video generation field (Xie et al., 2025; Yin et al., 2025; Tang et al., 2024; Zhou et al., 2025; Henschel et al., 2025; Zhai et al., 2025; Huang et al., 2025; Liu et al., 2025a; Deng et al., 2025a; Liu et al., 2025e; Ge et al., 2026; Xu et al., 2024). PA-VDM Xie et al. (2025) proposes a progressive noise-adding mechanism and denoises the frame using small intervals. CausVID Yin et al. (2025) adapts a pre-trained bidirectional transformer and designs a distribution matching distillation technique to reduce latency. Self Forcing Huang et al. (2025) presents the auto-regressive rollout with key-value (KV) caching during training. Recent studies have also emphasized temporal structure and long-horizon dependencies for dynamic-scene understanding (Liu et al., 2025f; 2024d; 2023d; Cheng et al., 2023b;a), motivating an auto-regressive formulation for optical flow. Inspired by the great success in these auto-regressive video generation works, we, for the first time, introduce the auto-regressive paradigm into optical flow estimation.

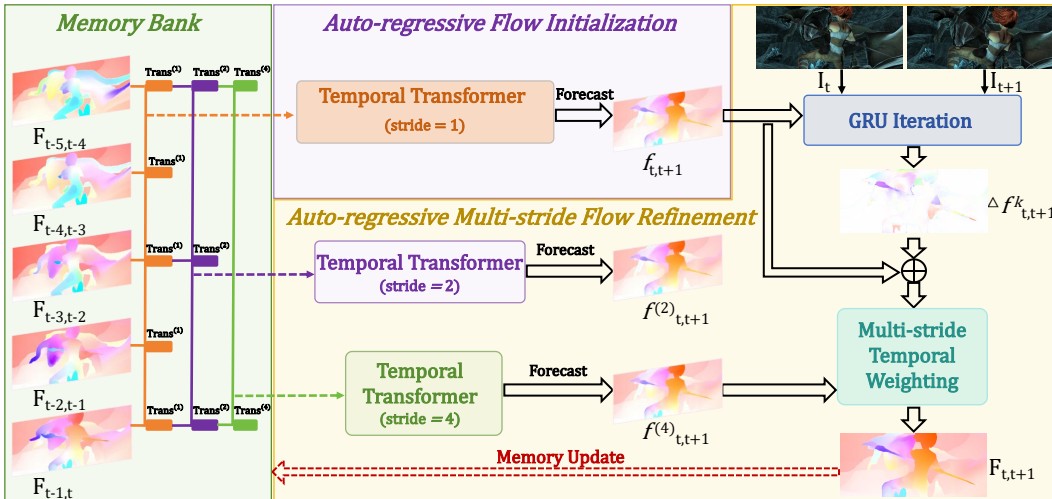

Figure 3: The overall architecture of our proposed ARFlow.

## 3 METHOD

### 3.1 OVERALL ARCHITECTURE

As in Fig. 1 (C), the input of our network is a video sequence $I_1, I_2, ..., I_{L-1}, I_L$, where $I_t \in H \times W \times 2$ indicates the image at the timestamp $t$. $H, W$ are the height and width of the input images. The outputs are corresponding sequential optical flows $F_{1,2}, F_{2,3}, ..., F_{L-1,L}$ between consecutive image pairs, where $F_{t,t+1}$ indicates the optical flow from $I_t$ to $I_{t+1}$.

In our auto-regressive setting, sequential images are delivered into the network frame by frame. When a new frame $I_{t+1}$ is inputted into Fig. 3, a forecasted flow initialization $f_{t,t+1}$ is first generated based on previously predicted $T$ frames' history flows stored in the memory bank in Section 3.2. Then, above initial flow $f_{t,t+1}$ is iteratively refined by incorporating intermediate context and motion features from $I_t$ to $I_{t+1}$ and multi-stride forecasted flows in Section 3.3. The output refined flow is finally used to update the memory bank and forecast the following flow $f_{t+1,t+2}$ in an auto-regressive manner. The network is supervised by the loss functions defined in Section 3.4.

### 3.2 AUTO-REGRESSIVE FLOW INITIALIZATION

Motivated by prior works from Xu et al. (2023a), long-range temporal dependencies are crucial for optical flow estimation, where potential challenges such as occlusions can be alleviated utilizing multi-frame temporal cues. In addition, optical flows for dynamic objects tend to be consistent across frames. Therefore, we use history flow estimates from a designed memory bank to forecast an extrapolated flow in the next frame for enhanced flow initialization.

**Memory Bank.** When a new pair of images ($I_t$ and $I_{t+1}$) is delivered to the network as in Fig. 3, our method first forecasts the intermediate initial flow based on historically predicted flows. To effectively store multiple history flows, we design a memory bank with a fixed length $T$, which stores the nearest $T$ frames' predicted flows $\{F_{i,i+1}\}_{i=t-T}^{t-1}$. When the number of predicted flows is less than $T$, the first several frames' flows are directly stored in the memory bank as memory initialization. The memory bank is maintained by a sliding window mechanism: When a new flow is predicted, it is stored in the memory bank, and the temporally-farthest one is discarded. To guarantee efficiency, our stored flows are at the 1/16 resolution of the original image size.

**Auto-regressive Next-Frame Forecasting.** For the intermediate optical flow estimation, we predict the next-frame flow based on previously stored flows as:

$$\{\mathbf{Feat}_i^{(1)}\}_{i=t-T}^{t-1} = \text{Trans}^{(1)}\left(\{F_{i,i+1}\}_{i=t-T}^{t-1}\right), \quad f_{t,t+1} = \phi(\mathbf{Feat}_{t-1}^{(1)}), \tag{1}$$

where $\text{Trans}^{(1)}$ is a transformer encoder for temporal modeling on all input frames, $\phi(\cdot)$ is a lightweight Conv2d projection, and $f_{t,t+1}$ denotes the initial flow predicted from the last token

$\mathbf{Feat}_{t-1}^{(1)}$. Since this initial flow predicted from multiple history flows possesses the long-range temporal cues, it can provide an enhanced flow initialization.

### 3.3 AUTO-REGRESSIVE MULTI-STRIDE FLOW REFINEMENT

After getting the initial flow $f_{t,t+1}$, we further leverage GRU-based iterations based on current observations and multi-stride history weighting to refine $f_{t,t+1}$. Notably, both refinements are conducted at the 1/16 of original resolution for fast inference.

**GRU-based Iterative Refinement.** We first leverage a GRU-based refinement to incorporate the information derived from current input images $I_t, I_{t+1}$. Specifically, the feature maps $E_t, E_{t+1}$ of current image pairs are first extracted by ResNet16 (He et al., 2016). Then, we obtain the intermediate context feature from the current images as:

$$c, h^0 = \text{ContextNetwork}(I_t, I_{t+1}), f^0 = \text{FlowHead}(h^0), \tag{2}$$

where ContextNetwork follows the setting in (Bargatin et al., 2025). $c$ and $h^0$ respectively indicate the context feature and the initial hidden state in GRU. $f^0$ is the predicted intermediate flow. If there are no stored history flows (the first image pairs), we use $f^0$ as the input of GRU. Otherwise, we use the temporally forecasted flow $f_{t,t+1}$ as the input flow in GRU as:

$$f_{t,t+1}^0 = \begin{cases} f^0, & when\ t = 0 ; \\ f_{t,t+1}, & otherwise , \end{cases} \tag{3}$$

where $f_{t,t+1}^0$ is the input initial flow of the GRU module. Also, we associate two frames to obtain a correlation feature map $V(u,v) = <E_t(u), E_{t+1}(v)>$, where $< \cdot, \cdot >$ denotes the dot product. $u, v$ denote the $u$-th and $v$-th element respectively in $E_t$ and $E_{t+1}$. For the $k$-th iteration, the correlation values are retrieved by the look-up operation from $V$ based on updated flow as: $V^k = \text{LookUp}(V, f_{t,t+1}^{k-1})$, where $f_{t,t+1}^{k-1}$ is the intermediate output flow from the $(k\text{-}1)$-th iteration. Then, the intermediate motion feature $M^k$ is generated by $M^k = \text{MotionEncoder}(\text{MLP}(V^k), \text{MLP}(f_{t,t+1}^{k-1}))$. Finally, the hidden state is iteratively updated by combining context feature $c$, motion feature $M^k$, and the previous hidden state $h^k = \text{GRU}(c, M^k, h^{k-1})$. Afterwards, the intermediate residual flows are obtained from the updated hidden state $\Delta f_{t,t+1}^k = \text{FlowHead}(h^k)$. The output flow after the $k$-th iteration is generated by per-point adding between the intermediate residual flow and initial flow as:

$$f_{t,t+1}^k = f_{t,t+1}^{k-1} + \Delta f_{t,t+1}^k. \tag{4}$$

We repeat the above process for $K$ times to generate the final refined flow $f_{t,t+1}^K$ after GRU.

**Multi-stride Temporal Weighting Refinement.** The above GRU-based iterations only incorporate the context and motion features from the current timestamp, without consideration of multi-stride history flows or motions. This leads to poor adaptation to various motion variations. To address the problem, we continually refine $f_{t,t+1}^K$ with multi-stride temporal forecasting.

Because the low-frequency and high-frequency information displays with unpredictable patterns along the entire video length, the single-stride temporal modeling in Section 3.2 is not effective enough to capture long-range motion variations. To enlarge the temporal receptive field and strengthen the adaptation ability to diverse motion variations, we also generate forecasted flows by the multi-stride temporal modeling in a cascaded manner. Specifically, we further sample frames with strides of 2 and 4 as:

$$\{\mathbf{Feat}_i^{(2)}\}_{i \in \{t-1,\,t-3,\,\dots\}} = \text{Trans}^{(2)}\Big(\{\mathbf{Feat}_i^{(1)}\}_{i \in \{t-1,\,t-3,\,\dots\}}\Big), \quad f_{t,t+1}^{(2)} = \phi(\mathbf{Feat}_{t-1}^{(2)}) \tag{5}$$

$$\{\mathbf{Feat}_i^{(4)}\}_{i \in \{t-1,\,t-5,\,\dots\}} = \text{Trans}^{(4)}\Big(\{\mathbf{Feat}_i^{(2)}\}_{i \in \{t-1,\,t-5,\,\dots\}}\Big), \quad f_{t,t+1}^{(4)} = \phi(\mathbf{Feat}_{t-1}^{(4)}) \tag{6}$$

$$\{\mathbf{Feat}_{\text{fuse}}^{(l)}\}_{l \in \{1,\,2,\,4\}} = \text{Trans}^{(f)}\Big(\mathbf{Feat}_{t-1}^{(1)}, \mathbf{Feat}_{t-1}^{(2)}, \mathbf{Feat}_{t-1}^{(4)}\Big), \quad f_{\text{fuse}} = \phi(\mathbf{Feat}_{\text{fuse}}^{(4)}), \tag{7}$$

where $\text{Trans}^{(2)}$ and $\text{Trans}^{(4)}$ perform temporal modeling with stride 2 and stride 4, and $\text{Trans}^{(f)}$ aggregates multi-stride features. Both $f_{t,t+1}^{(2)}$ and $f_{t,t+1}^{(4)}$ are used as auxiliary supervision in the

Table 1: **Benchmark results on MPI-Sintel and KITTI-15.** We report endpoint-error (EPE) on Sintel (Butler et al., 2012) and Fl on KITTI-15 (Geiger et al., 2013).

| | Method | Reference | Sintel Clean | | | Sintel Final | | | KITTI-15 | |
|---|---|---|---|---|---|---|---|---|---|---|
| | | | Mat.↓ | Unm.↓ | All↓ | Mat.↓ | Unm.↓ | All↓ | All↓ | Non-Occ↓ |
| Pairwise | FlowNet2 (Ilg et al., 2017) | CVPR'17 | 1.56 | 25.40 | 4.16 | 2.75 | 30.11 | 5.74 | 10.41 | 6.94 |
| | PWC-Net (Sun et al., 2018) | CVPR'18 | 1.45 | 23.47 | 3.86 | 2.44 | 27.08 | 5.04 | 9.60 | 6.12 |
| | RAFT (Teed & Deng, 2020) | ECCV'20 | 0.62 | 9.65 | 1.61 | 1.41 | 14.68 | 2.86 | 5.10 | 3.07 |
| | GMFlow (Xu et al., 2022) | CVPR'22 | 0.65 | 10.56 | 1.74 | 1.32 | 15.80 | 2.90 | 9.32 | 3.80 |
| | FlowFormer (Huang et al., 2022) | CVPR'22 | 0.42 | 7.16 | 1.16 | 0.96 | 11.30 | 2.09 | 4.68 | 2.69 |
| | GMFlow+ (Xu et al., 2023b) | TPAMI'23 | 0.34 | 6.68 | 1.03 | 1.10 | 12.74 | 2.37 | 4.49 | 2.40 |
| | Flowformer++ (Shi et al., 2023b) | CVPR'23 | 0.39 | 6.64 | 1.07 | 0.88 | 10.63 | 1.94 | 4.52 | - |
| | AnyFlow (Jung et al., 2023) | CVPR'23 | 0.42 | 7.68 | 1.21 | 1.12 | 13.37 | 2.44 | 4.41 | 2.69 |
| | CroCoFlow (Weinzaepfel et al., 2023) | ICCV'23 | 0.39 | 6.85 | 1.09 | 1.21 | 12.42 | 2.44 | 3.64 | 2.40 |
| | DDVM (Saxena et al., 2023) | NeurIPS'23 | 0.83 | 9.26 | 1.75 | 1.28 | 12.20 | 2.48 | 3.26 | 2.24 |
| | FlowDiffuser (Luo et al., 2024) | CVPR'24 | 0.38 | 6.23 | 1.02 | 0.97 | 10.67 | 2.03 | 4.17 | 2.82 |
| | SEA-RAFT(L) (Wang et al., 2024b) | ECCV'24 | 0.44 | 8.40 | 1.31 | 1.20 | 14.06 | 2.60 | 4.30 | - |
| | SAMFlow (Zhou et al., 2024) | AAAI'24 | 0.38 | 5.97 | 1.00 | 1.04 | 10.60 | 2.08 | 4.49 | - |
| | DPFlow (Morimitsu et al., 2025) | CVPR'25 | 0.39 | 6.36 | 1.04 | 0.91 | 10.69 | 1.97 | 3.56 | 2.12 |
| | CEDFlow++ (Zuo et al., 2025) | IJCV'25 | - | - | 1.37 | - | - | 2.40 | 4.78 | - |
| | WAFT (Wang & Deng, 2025) | Arxiv'25 | - | - | 1.09 | - | - | 2.34 | 3.42 | 2.04 |
| Multi-frame | MFCFlow (Chen et al., 2023) | WACV'23 | 0.65 | 8.34 | 1.49 | 1.33 | 12.81 | 2.58 | 5.00 | - |
| | TransFlow (Lu et al., 2023) | CVPR'23 | 0.36 | 6.77 | 1.06 | 0.99 | 10.96 | 2.08 | 4.32 | - |
| | SplatFlow (Wang et al., 2024a) | IJCV'24 | 0.51 | 6.06 | 1.12 | 1.06 | 10.29 | 2.07 | 4.61 | 2.96 |
| | StreamFlow (Sun et al., 2024) | NeurIPS'24 | 0.38 | 6.42 | 1.04 | 0.82 | 10.44 | 1.87 | 4.24 | 2.45 |
| | MemFlow (Dong & Fu, 2024) | CVPR'24 | 0.43 | 6.09 | 1.05 | 0.93 | 9.93 | 1.91 | 4.10 | 2.56 |
| | MEMFOF (Bargatin et al., 2025) | ICCV'25 | 0.40 | 5.59 | 0.96 | 0.88 | 10.30 | 1.91 | 2.94 | 1.97 |
| | **ARFlow (Ours)** | — | 0.39 | 5.64 | 0.96 | 0.81 | 9.79 | 1.78 | 2.85 | 1.91 |

loss function to stabilize training and guide the learning of $\text{Trans}^{(2)}$ and $\text{Trans}^{(4)}$. From the final token of $\text{Trans}^{(f)}$, we additionally predict a learnable weighting parameter $w_{t,t+1} = \phi_f(\mathbf{Feat}^{(4)}_{\text{fuse}})$. Finally, we re-weight the output from the GRU refinement by incorporating multi-stride forecasted flows by:

$$F_{t,t+1} = w_{t,t+1}f^K_{t,t+1} + (1 - w_{t,t+1})f_{\text{fuse}}. \tag{8}$$

The weighted optical flow estimation $F_{t,t+1}$ is the final output, which is also leveraged to update the memory bank to forecast the next frame pairs. The final flow estimation $F_{t,t+1}$ are convexly upsampled to the input resolution as in RAFT (Teed & Deng, 2020).

## 3.4 LOSS FUNCTIONS

Following SEA-RAFT (Wang et al., 2024b), we utilize a mixture-of-Laplace (MoL) loss and apply RAFT-style deep supervision across iterations. Given $T$ frames and $K$ refinement steps, the training loss is

$$\mathcal{L} = \frac{1}{T}\sum_{t=1}^{T}\sum_{k=0}^{K}\gamma^{K-k}\mathcal{L}^{t,k}_{\text{MoL}}, \tag{9}$$

where $\mathcal{L}^{t,k}_{\text{MoL}}$ denotes the MoL loss for frame $t$ at refinement $k$. Following (Bargatin et al., 2025), we set $\gamma = 0.85$ to emphasize later refinements. Parameterization and mixture details are provided in the supplementary. In addition to $K$ refined flows $f^k_{t,t+1}$, we also apply the same MoL objective to all intermediate flows, including $f_{t,t+1}$, $f^{(2)}_{t,t+1}$, $f^{(4)}_{t,t+1}$, $f_{\text{fuse}}$, and $F_{t,t+1}$.

## 4 EXPERIMENTS

### 4.1 EXPERIMENTAL SETTINGS

**Evaluation Datasets and Metrics.** We evaluate on three standard optical-flow benchmarks: Spring (Mehl et al., 2023b) (modern high-resolution video), MPI-Sintel (Butler et al., 2012) (synthetic scenes with complex motion), and KITTI-2015 (Menze & Geiger, 2015) (autonomous driving). We report endpoint error (EPE), 1-pixel error rate (1px), Fl (KITTI-15), and WAUC as comparison metrics. The detailed definitions are described in the appendix.

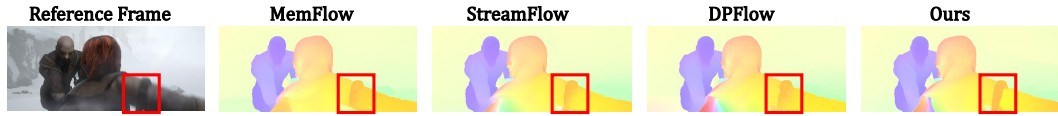

Figure 4: Qualitative comparison on Sintel test set. Compared with previous methods, our ARFlow can predict sharper motion boundaries as highlighted in the circle. Please zoom in for details.

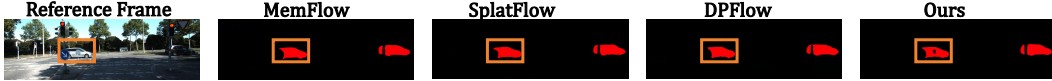

Figure 5: Qualitative comparison on KITTI test set. Compared with previous methods, our ARFlow can distinguish occluded areas as highlighted in the circle. Please zoom in for details.

**Network Architectures.** For image feature extraction, we use a ResNet–FPN backbone He et al. (2016) (ResNet-34, dim= 512) shared by the context network (cnet) and feature network (fnet). cnet takes concatenated RGB frames to produce 1/16-resolution features and the initial hidden state, while fnet extracts per-frame features for subsequent feature matching. On top of these encoders, we adopt a standard RAFT/GMA-style backbone (Jiang et al., 2021) with a 4-stage correlation pyramid (radius=4), a GMAUpdateBlock (num_blocks= 2, iters= 6 at 1/16 resolution), and a learned convex upsampler for GRU-based Iterative Refinement. For the Temporal Transformer, we first encode the optical flow and its uncertainty at each time step using 2D convolutions, transforming the input $(B, T, 6, H, W)$ into features $(B, T, C, H, W)$. We then rearrange them as: $(B, T, C, H, W) \rightarrow (BHW, T, C)$, and apply standard multi-head self-attention along the temporal dimension. Moreover, we adopt multi-scale hierarchical Temporal Transformers (downsampled by the strides=1, 2, 4 over time) and fuse these sequences with another Temporal Transformer in the Auto-regressive Multi-Stride Flow Refinement module.

**Pretraining and Fine-Tuning for Benchmark Submissions.** We first pretrain on TartanAir (Wang et al., 2020) for 225k steps with a crop size of $480 \times 960$, batch size of 64, and a learning rate of $1.4 \times 10^{-4}$, which requires about 13.1 GB of GPU memory. This model is then used to initialize training on FlyingThings3D (Mayer et al., 2016) for 120k steps with a larger crop size of $864 \times 1920$, batch size 32, and learning rate $7 \times 10^{-5}$ (18.5 GB). Next, we extend the training to the combined T+S+K+H (FlyingThings (Mayer et al., 2016), Sintel (train) (Butler et al., 2012), KITTI-15 (train) (Geiger et al., 2013), and HD1K (Kondermann et al., 2016)) set for 225k steps under the same settings.

For benchmark-specific adaptation, we perform lightweight fine-tuning. On Sintel, we train 12.5k steps with $872 \times 1920$ crops, batch size 8, and learning rate $3 \times 10^{-5}$, requiring 23.8 GB of memory. On KITTI-15, we fine-tune 2.5k steps at $750 \times 1920$, batch size 32, and learning rate $3 \times 10^{-5}$, requiring 20.8 GB of memory. On Spring (Mehl et al., 2023b), we run 60k steps with full-resolution $1080 \times 1920$ crops, batch size 8, and learning rate $4.8 \times 10^{-5}$, requiring 29.4 GB of memory. We adopt MEMFOF (Bargatin et al., 2025) as the network architecture. By default, the temporal length of the memory bank is $T = 6$, and the GRU refinement performs $K = 6$ iterations. The detailed network implementation is provided in the supplementary code.

**Settings for Zero-Shot Evaluation.** For zero-shot evaluation, following MemFlow (Dong & Fu, 2024), we first pre-train the networks in the 2-frame setting on FlyingChairs (60k iterations) and FlyingThings3D (150k iterations). We then enable temporal modeling and continue training on FlyingThings3D for an additional 120k iterations.

## 4.2 BENCHMARK RESULTS

**Benchmark Results on Sintel and KITTI-15.** Table 1 reports results on the two widely-used benchmarks. **ARFlow** ranks **first** on the official KITTI-15 benchmark (Geiger et al., 2013), achieving the lowest errors on both *All* pixels (**2.85** vs. 2.94 of MEMFOF and 4.10 of MemFlow) and *Non-Occ* pixels (**1.91** vs. 1.97 of MEMFOF and 2.56 of MemFlow). On Sintel (Butler et al., 2012), ARFlow achieves state-of-the-art performance on the *Final* pass (**1.78** vs. 1.87 of StreamFlow and 1.91 of MEMFOF) and ties the best on *Clean* (0.96). These results highlight the advantage of our auto-regressive framework: *(AFI)* provides reliable initialization values from historical estimates, while *(AMFR)* integrates both short- and long-term temporal cues, enabling more accurate

Table 2: **Benchmark results on Spring.** Runtime and maximum GPU memory usage were evaluated using an NVIDIA RTX 3090 GPU. Best results are respectively highlighted as **first**, second. OOM indicates out of memory. * indicates scene flow methods.

| | Method | #Frames | Inference Cost (1080p) | | Spring (test) | | | |
|---|---|---|---|---|---|---|---|---|
| | | | Memory, GB | Runtime, ms | 1px ↓ | EPE ↓ | Fl ↓ | WAUC ↑ |
| NO FINE-TUNE | Flow1D (Xu et al., 2021) | 2 | 1.34 | 405 | - | - | - | - |
| | MeFlow (Xu et al., 2023a) | 2 | 1.32 | 1028 | - | - | - | - |
| | PWC-Net (Sun et al., 2018) | 2 | 1.41 | 76 | 82.265 | 2.288 | 4.889 | 45.670 |
| | FlowNet2 (Ilg et al., 2017) | 2 | 4.16 | 167 | 6.710 | 1.040 | 2.823 | 90.907 |
| | RAFT (Teed & Deng, 2020) | 2 | 7.97 | 557 | 6.790 | 1.476 | 3.198 | 90.920 |
| | RAFT3D* (Teed & Deng, 2021) | 2 | - | - | 13.962 | 2.528 | 6.889 | 81.267 |
| | GMA (Jiang et al., 2021) | 2 | 13.26 | 1185 | 7.074 | 0.914 | 3.079 | 90.722 |
| | GMFlow (Xu et al., 2022) | 2 | - | 151 | 10.355 | 0.945 | 2.952 | 82.337 |
| | FlowFormer (Huang et al., 2022) | 2 | OOM | - | 6.510 | 0.723 | 2.384 | 91.679 |
| | RPKNet (Morimitsu et al., 2024) | 2 | 8.49 | 295 | 4.809 | 0.657 | 1.756 | 92.638 |
| | Win-Win (Leroy et al., 2024) | 2 | - | - | 5.371 | 0.475 | 1.621 | 92.720 |
| | MS-RAFT+ (Jahedi et al., 2024) | 2 | - | - | 5.724 | 0.643 | 2.189 | 92.888 |
| | MatchAttention (Yan et al., 2025) | 2 | 14.34 | 755 | 4.584 | 0.453 | 1.505 | 93.389 |
| | M-Fuse (F)* (Mehl et al., 2023a) | 3 | - | - | 20.374 | 2.948 | 8.791 | 76.550 |
| | VideoFlow-BOF (Shi et al., 2023a) | 3 | 17.74 | 1648 | - | - | - | - |
| | VideoFlow-MOF (Shi et al., 2023a) | 5 | OOM | - | - | - | - | - |
| | MemFlow (Dong & Fu, 2024) | 3 | 8.08 | 885 | 5.759 | 0.627 | 2.114 | 92.253 |
| | StreamFlow (Sun et al., 2024) | 4 | 18.97 | 929 | 5.215 | 0.606 | 1.856 | 93.253 |
| | MEMFOF (Bargatin et al., 2025) | 3 | 2.09 | 472 | 3.600 | 0.432 | 1.353 | 94.481 |
| | **ARFlow (Ours)** | 8 | 2.10 | 403 | **3.587** | **0.428** | **1.313** | **94.501** |
| FINE-TUNE | CrocoFlow (Weinzaepfel et al., 2023) | 2 | 2.01 | 6524 | 4.565 | 0.498 | 1.508 | 93.660 |
| | SEA-RAFT (S) (Wang et al., 2024b) | 2 | 8.15 | 205 | 3.904 | 0.377 | 1.389 | 94.182 |
| | SEA-RAFT (M) (Wang et al., 2024b) | 2 | 8.19 | 286 | 3.686 | 0.363 | 1.347 | 94.534 |
| | MemFlow (Dong & Fu, 2024) | 3 | 8.08 | 885 | 4.482 | 0.471 | 1.416 | 93.855 |
| | StreamFlow (Sun et al., 2024) | 4 | 18.97 | 929 | 4.152 | 0.467 | 1.424 | 94.404 |
| | DPFlow (Morimitsu et al., 2025) | 2 | 10.39 | 990 | 3.442 | **0.340** | 1.280 | 94.663 |
| | MEMFOF (Bargatin et al., 2025) | 3 | 2.09 | 472 | 3.289 | 0.355 | 1.238 | 95.186 |
| | **ARFlow (Ours)** | 8 | 2.10 | 403 | **3.265** | 0.353 | **1.212** | **95.283** |

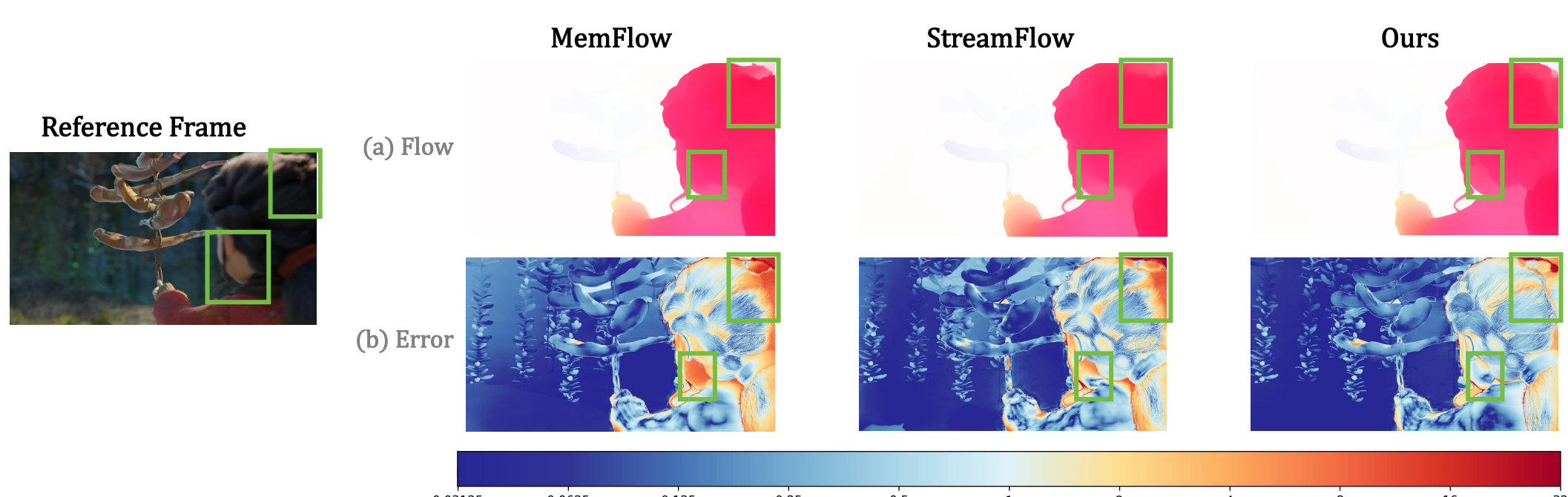

Figure 6: Qualitative comparison on Spring test set. Compared with previous multi-frame methods MemFlow (Dong & Fu, 2024), StreamFlow (Sun et al., 2024), our ARFlow has more accurate flow estimations and lower errors as highlighted in the circle. Please zoom in for details.

and robust predictions. We also visualize the qualitative flow predictions as in Fig. 4 and Fig. 5, respectively. Our ARFlow performs better in terms of sharp boundaries and occluded areas.

**Benchmark Results on Spring.** Table 2 shows the results on Spring. With full-resolution training (1080p), ARFlow achieves the state-of-the-art performance across all four metrics (1px, EPE, Fl, WAUC), while being highly efficient in both memory (2.1 GB) and runtime. This efficiency stems from avoiding redundant feature extraction and progressively forecasting frame by frame, making ARFlow the only method that combines SOTA accuracy with scalability to long sequences. We also visualize the predicted flow and its corresponding error maps in Fig. 6.

Table 3: **Zero-shot Generalization.** ARFlow achieves the best cross-dataset generalization on KITTI-15 (train).

| Method | Sintel | | KITTI-15 | |
|---|---|---|---|---|
| | Clean ↓ | Final ↓ | Fl-EPE ↓ | Fl-all ↓ |
| PWC-Net (Sun et al., 2018) | 2.55 | 3.93 | 10.40 | 33.7 |
| FlowNet2 (Ilg et al., 2017) | 2.02 | 3.14 | 10.10 | 30.4 |
| RAFT (Teed & Deng, 2020) | 1.43 | 2.71 | 5.04 | 17.4 |
| SKFlow (Sun et al., 2022) | 1.22 | 2.46 | 4.27 | 15.5 |
| GMFlowNet (Zhao et al., 2022) | 1.14 | 2.71 | 4.24 | 15.4 |
| FlowFormer (Huang et al., 2022) | 1.01 | 2.40 | 4.09 | 14.7 |
| SEA-RAFT(L) (Wang et al., 2024b) | 1.19 | 4.11 | 3.62 | 12.9 |
| AnyFlow (Jung et al., 2023) | 1.10 | 2.52 | 3.76 | 12.4 |
| FlowDiffuser (Luo et al., 2024) | 0.86 | 2.19 | 3.61 | 11.8 |
| DPFlow (Morimitsu et al., 2025) | 1.02 | 2.26 | 3.37 | 11.1 |
| FlowSeek (Poggi & Tosi, 2025) | 1.03 | 2.18 | 3.31 | 11.2 |
| WAFT (Wang & Deng, 2025) | 1.00 | 2.15 | 3.10 | 10.3 |
| MEMFOF (Bargatin et al., 2025) | 1.20 | 3.91 | 2.93 | 9.9 |
| **ARFlow (Ours)** | 0.88 | 2.07 | 2.86 | 9.2 |

Table 4: **Compatibility evaluation on various baselines.** Consistent with (Wang et al., 2024b; Sun et al., 2022; Morimitsu et al., 2025), all models are trained on the combined Clean+Final (C+T) split and evaluated on the Sintel and KITTI-2015 training sets for a fair comparison. "PG." indicates the performance gain over the baseline.

| Method | Sintel (train) | | | KITTI-2015 (train) | | |
|---|---|---|---|---|---|---|
| | Clean ↓ | Final ↓ | PG. | EPE ↓ | F1-all ↓ | PG. |
| SEA-RAFT(L) (Wang et al., 2024b) | 1.19 | 4.11 | – | 3.62 | 12.9 | – |
| ARFlow-S | **1.12** | **3.72** | 9.5% ↑ | **3.32** | **11.4** | 11.6% ↑ |
| DPFlow (Morimitsu et al., 2025) | 1.02 | 2.26 | – | 3.37 | 11.1 | – |
| ARFlow-D | **0.97** | **2.14** | 5.3% ↑ | **3.08** | **10.3** | 7.2% ↑ |
| FlowSeek (Poggi & Tosi, 2025) | 1.03 | 2.18 | – | 3.31 | 11.2 | – |
| ARFlow-F | **0.95** | **2.07** | 5.0% ↑ | **3.05** | **10.1** | 9.8% ↑ |
| WAFT (Wang & Deng, 2025) | 1.00 | 2.15 | – | 3.10 | 10.3 | – |
| ARFlow-W | **0.91** | **2.01** | 6.5% ↑ | **2.93** | **9.4** | 8.7% ↑ |

## 4.3 ZERO-SHOT GENERALIZATION

Following common practice from Teed & Deng (2020); Huang et al. (2022); Dong & Fu (2024), we pretrain our model on the FlyingChairs (Dosovitskiy et al., 2015) and FlyingThings3D (Mayer et al., 2016) datasets, and then directly assess its performance on the training splits of Sintel (Butler et al., 2012) and KITTI-15 (Geiger et al., 2013).Table 3 summarizes cross-dataset generalization performance. ARFlow achieves strong transfer performance across Sintel (train) and KITTI-15 (train), consistently outperforming recent baselines. These results demonstrate that our auto-regressive design not only improves benchmark accuracy but also yields robust generalization to unseen datasets.

## 4.4 COMPATIBILITY EVALUATION ON OTHER BASELINES

As shown in Table 4, ARFlow consistently improves different baselines across Sintel and KITTI-2015. For example, it yields 9.5% and 11.6% gains when integrated with SEA-RAFT, and also achieves 5–10% relative improvements when combined with the SOTA methods DPFlow, FlowSeek, and WAFT. These results indicate that our auto-regressive initialization and multi-stride refinement provide complementary temporal cues, serving as a general plug-in that enhances both transformer- and pyramid-based architectures.

## 4.5 ABLATION STUDY

**Effectiveness of AFI and AMFR.** Table 5 (#1) evaluates the influence of the proposed components. Removing the Auto-regressive Flow Initialization (AFI) increases errors notably, e.g., Sintel (Clean/Final) from 0.88/2.07 to 0.95/2.15, and KITTI-2015 EPE rises from 2.86 to 3.05. Similarly, discarding the Auto-regressive Multi-stride Flow Refinement (AMFR) degrades performance (Sintel 0.91/2.13, KITTI-2015 EPE 3.00). These results confirm that both AFI and AMFR are indispensable for accurate and robust estimation.

**Stride Design in AMFR.** As shown in Table 5 (#2), using only a single stride (1, 2, or 4) yields weaker performance. Combining multi-stride settings achieves the lowest errors (Sintel 0.88/2.07, KITTI-2015 EPE 2.86) compared

Table 5: **Ablation study.** Settings used as default are underlined. All models are trained on C+T for fair comparison.

| Method | Sintel (train) | | KITTI-2015 (train) | |
|---|---|---|---|---|
| | Clean ↓ | Final ↓ | EPE ↓ | F1-all ↓ |
| #1: ARFlow (AF.) approaches | | | | |
| AF.-w/o AFI | 0.95 | 2.15 | 3.05 | 10.11 |
| AF.-w/o AMFR | 0.91 | 2.13 | 3.00 | 9.73 |
| AF. | **0.88** | **2.07** | **2.86** | **9.21** |
| #2: AMFR Strategy | | | | |
| Single Stride 1 | 0.89 | 2.10 | 2.89 | 9.52 |
| Single Stride 2 | 0.90 | 2.12 | 2.93 | 9.55 |
| Single Stride 4 | 0.91 | 2.12 | 3.00 | 9.68 |
| Stride 1 + 2 + 4 | **0.88** | **2.07** | **2.86** | **9.21** |
| #3: Temporal Length $T$ | | | | |
| 4 | 0.91 | 2.11 | 2.96 | 9.43 |
| 5 | 0.90 | 2.09 | 2.92 | 9.41 |
| 6 | **0.88** | 2.07 | **2.86** | 9.21 |
| 7 | **0.88** | 2.08 | 2.88 | 9.19 |
| 8 | 0.89 | **2.06** | **2.86** | **9.17** |

to any of the single stride settings, demonstrating the benefit of capturing both short- and long-term motions.

**Temporal Length.** Table 5 (#3) investigates the impact of varying the memory length $T$. We observe consistent gains when extending from 4 to 6 frames, with the best results at $T = 6$ (KITTI-2015 EPE 2.86, Sintel 0.88/2.07). Further increasing frames shows a marginal change, suggesting a trade-off between accuracy and efficiency.

## 4.6 DISCUSSION

**Attention Weights in Temporal Modeling Transformer.** We also visualize the attention weights in our designed auto-regressive transformer module in Fig. 7. We observe that our transformer mainly focuses on dynamic objects and motion boundaries, which can provide an accurate motion prior for the next-frame flow forecasting. This ensures a stable and robust flow initialization, resulting in excellent performance of our ARFlow.

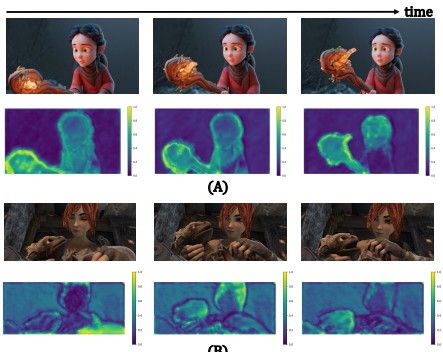

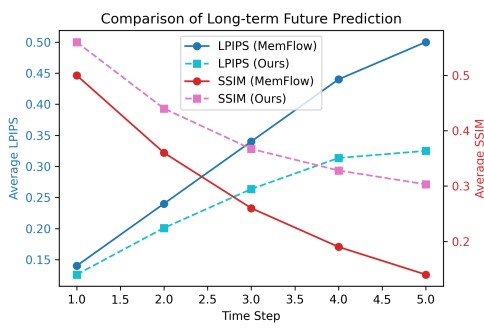

Figure 7: Visualization of attention weights in our temporal modeling transformer.

Figure 8: Quantitative comparison of long-term future prediction by optical flow.

**Downstream Task: Video Prediction.** We evaluate ARFlow against MemFlow under different prediction horizons using LPIPS (lower is better) (Zhang et al., 2018) and SSIM (higher is better) (Wang et al., 2004). As in Fig. 8, when the prediction horizon increases, MemFlow exhibits a rapid degradation in performance, while ARFlow demonstrates a much slower decline, highlighting its stronger temporal modeling and autoregressive prediction capability. Unlike MemFlow, which requires an additional prediction head, ARFlow naturally predicts the next-frame optical flow, leading to more stable and consistent performance over longer horizons. Additional implementation details are provided in the supplementary material.

## 5 CONCLUSION

In this paper, we propose a novel multi-frame optical flow estimation pipeline based on auto-regressive transformer models. We formulate the task with the progressive flow estimation frame by frame and introduce an auto-regressive next-frame forecasting strategy to provide accurate and robust initial flow. Then, a refinement module combining both current motion features and multi-stride history forecasting flows is leveraged to generate fine-grained flow estimates and refine the initial predictions. Extensive experiments on standard benchmarks demonstrate that our method achieves state-of-the-art accuracy while maintaining competitive efficiency.

ACKNOWLEDGEMENTS

This work was supported by National Key R&D Program of China (Grant No.2024YFB4708900) and EU HORIZON-CL4-2023-HUMAN-01-CNECT XTREME (Grant No. 101136006). It was also supported in part by the Natural Science Foundation of China under Grant 62225309,U24A20278, 62361166632.

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

APPENDIX

## A  OVERVIEW

The supplementary materials are structured as follows:

- We give more detailed illustrations about the metrics, loss functions, settings in video prediction, and clip-wise training strategy in Section B;
- More experimental results are provided in Section C.
- Qualitative results on KITTI, Waymo, and NuScenes are presented in Section D;
- Screenshots for benchmark results on KITTI, Sintel, and Spring are shown in Section E;
- Section F discloses the limited and strictly assistive usage of a large language model (LLM) during manuscript polishing.
- A video demo of both synthetic and real-world driving scenes is appended to the supplementary materials, and the detailed network implementation is also provided in the supplementary code.

## B  IMPLEMENTATION DETAILS

### B.1  DEFINITION OF METRICS

We provide more formal definitions of metrics used in the main text.

- EPE is the average per-pixel Euclidean distance between the predicted flow and the reference flow over all valid pixels.
- The 1px rate is the fraction of valid pixels whose flow error exceeds one pixel.
- The Fl score (KITTI-2015) is the percentage of pixels whose error is larger than three pixels and also greater than five percent of the ground-truth magnitude.
- WAUC summarizes performance by integrating the inlier rate over error thresholds from zero to five pixels, with larger weights near zero; a precise definition appears in the supplementary material. For optical flow, the weighted area under the inlier-rate curve (WAUC), introduced with VIPER (Richter et al., 2017), aggregates the inlier percentage over error thresholds.

Let $f(\tau) \in [0, 100]$ denote the percentage of pixels whose endpoint error is at most $\tau$ pixels. With a weight that emphasizes small thresholds, the metric is

$$\text{WAUC} \;=\; \frac{2}{5} \int_0^5 f(\tau)\,\frac{5-\tau}{5}\,d\tau, \tag{10}$$

yielding a score between 0 (worst) and 100 (best).

### B.2  MIXTURE-OF-LAPLACE LOSS

Following SEA-RAFT, we use a two-component Laplace mixture for each scalar flow coordinate. Given target $y$ and predicted mean $\mu$, mixture weight $\alpha \in (0,1)$, and log-scale $\beta \in \mathbb{R}$, the per-coordinate negative log-likelihood is

$$\ell_{\text{mixlap}}(y; \alpha, \beta, \mu) = -\log\left[\frac{\alpha}{2}\,e^{-|y-\mu|} \;+\; \frac{1-\alpha}{2e^{\beta}}\,\exp\left(-\frac{|y-\mu|}{e^{\beta}}\right)\right]. \tag{11}$$

For a single optical-flow prediction, the image-level loss sums over spatial locations and the two flow coordinates:

$$\mathcal{L}_{\text{MoL}} = \frac{1}{2HW} \sum_{h=1}^{H} \sum_{w=1}^{W} \sum_{d \in \{x,y\}} \ell_{\text{mixlap}}\big(y_{h,w}^{(d)};\, \alpha_{h,w}, \beta_{h,w}, \mu_{h,w}^{(d)}\big). \tag{12}$$

All parameters $(\alpha, \beta, \mu)$ are predicted per pixel.

### B.3 SETTINGS FOR VIDEO PREDICTION.

For a fair comparison, we follow MemFlow (Dong & Fu, 2024): given the last frame $\mathbf{I}_t$, we first predict the forward flow $f_{t\rightarrow t+1}$ and a monocular depth map using DPT (Ranftl et al., 2021). We then render a candidate next frame by forward-warping $\mathbf{I}_t$ with Softmax Splatting (Niklaus & Liu, 2020), which yields a splatted image and a disocclusion mask identifying unsupported pixels. Finally, we complete those regions via ZITS inpainting (Dong et al., 2022), producing the synthesized frame $\hat{\mathbf{I}}_{t+1}$.

### B.4 CLIP-WISE TRAINING STRATEGY.

Compared to prior mainstream methods, the biggest difference is that our ARFlow keeps the original temporal sequence during the training process. However, most of the previous multi-frame methods, such as MemFlow Dong & Fu (2024) and StreamFlow Sun et al. (2024), interrupt the original temporal sequence by the random training batch shuffle. To be specific, previous methods like MemFlow use the standard batch-wise training strategy, where timestamps among consecutive input batches are not continuous. Thus, their temporal modeling remains constrained to these segments, like 3 frames within one batch, rather than truly covering arbitrary-length sequences because of the shuffled timestamps. This train–test inconsistency cannot reasonably be regarded as effective temporal modeling for long-sequence videos. Furthermore, if the temporal range in one batch is increased, they would suffer from infeasible computational burdens as in Figure 2 of the main manuscript.

In contrast, we treat the whole sequential video clip as input and use a clip-wise training method inspired by MOTR Zeng et al. (2022), rather than the batch-based one. In this case, our memory and supervision can model the temporal information for the whole sequence by sliding along all the frames. Therefore, our temporal receptive field can be viewed as the whole-sequence awareness for each video.

## C ADDITIONAL EXPERIMENTS

We also supplement more ablation studies and prediction results here.

**Temporal Flow Memory Resolution.** We ablate the resolution at which cached flows are stored, comparing $1\times$, $1/4\times$, and $1/16\times$. All models are trained on C+T for fair comparison. As shown in Table 6, storing flows at $1/16\times$ maintains accuracy on Sintel and KITTI while substantially reducing memory. We therefore adopt $1/16\times$ as the default setting in all experiments.

**Number of Iterative Refinements (K).** Following MEMFOF Bargatin et al. (2025), we vary the number of GRU refinement iterations $K\in\{4,5,6,7,8\}$ while keeping training on C+T for fairness. As reported in Table 6 (#2), increasing $K$ from 4 to 6 consistently improves Sintel and KITTI metrics. However, increasing to $K=7$ or $K=8$ yields mixed yet marginal changes (some metrics slightly improve while others slightly degrade), with overall gains being negligible relative to $K=6$ and accompanied by higher computation and memory costs. For a balance between speed and accuracy, we choose to perform 6 iterative refinements. For reference, MEMFOF reports using $K=8$ in their final configuration; since our pipeline builds on MEMFOF as the baseline but benefits from stronger autoregressive initialization and multi-stride forecast fusion, fewer refinement steps suffice in practice.

**Pre-trained Temporal Transformer.** We replace our Temporal Transformer from our original network with two recent pre-trained Transformer backbones: WAN2.2 (5B) Wan et al. (2025) and Longcat-video (14B) Team et al. (2025). As reported in Table 6 (#3), adding recent pre-trained transformers cannot bring additional performance gains, and our designed temporal transformer already has effective temporal modeling ability.

**Sub-sequence Results on Spring Dataset.** All the 11 sub-class 1px values on Spring benchmark are listed in Table 7. , where our ARFlow surpasses all recent state-of-the-art methods on most metrics, which demonstrates the excellent performance of our proposed method.

**Optical Flow Prediction.** We also evaluate the performance in flow prediction in Table 8. Compared to previous methods, ARFlow also achieves the best accuracy in the flow prediction task,

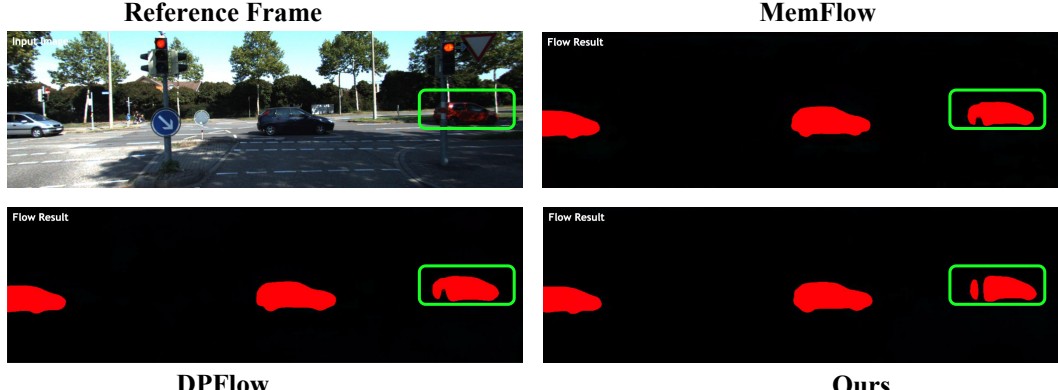

Figure 9: Qualitative results of MemFlow, DPFlow, and our method on the KITTI benchmark with occlusions. Sourced from official leaderboard submissions.

outperforming the recent method MemFlow-P by 10.2%, 13.9%, and 17.8%, respectively, on three datasets. We attribute this great prediction performance to our auto-regressive paradigm, which naturally captures long-range temporal cues and facilitates the flow extrapolation in a frame-by-frame auto-regression manner.

Table 6: **Additional Ablation study.** Settings used as default are underlined. All models are trained on C+T for fair comparison.

| Method | Sintel (train) | | KITTI-2015 (train) | |
|---|---|---|---|---|
| | Clean↓ | Final↓ | EPE↓ | F1-all↓ |
| #1: Temporal Flow Memory Resolution | | | | |
| 1 | 0.90 | 2.06 | 2.87 | 9.27 |
| 1/4 | 0.89 | 2.09 | 2.85 | 9.31 |
| 1/16 | 0.88 | 2.07 | 2.86 | 9.21 |
| #2: The number of iterative refinements (K). | | | | |
| 4 | 0.90 | 2.11 | 2.98 | 9.89 |
| 5 | 0.88 | 2.10 | 2.87 | 9.35 |
| 6 | 0.88 | 2.07 | 2.86 | 9.21 |
| 7 | 0.88 | 2.08 | 2.87 | 9.20 |
| 8 | 0.89 | 2.09 | 2.85 | 9.23 |
| #3: Pre-trained Transformer | | | | |
| WAN2.2 (5B) Wan et al. (2025) | 0.90 | 2.08 | 2.88 | 9.56 |
| Longcat-video (14B) Team et al. (2025) | 0.91 | **2.07** | 2.89 | 9.43 |
| w/o pre-trained Transformers | **0.88** | **2.07** | **2.86** | **9.21** |

## D    QUALITATIVE RESULTS ON KITTI, WAYMO AND NUSCENES

We provide additional qualitative comparisons on KITTI, Waymo, and NuScenes datasets.

**Visualizations on Occlusions.** On KITTI test sequences, our ARFlow produces sharper motion boundaries and better handles occluded regions compared with recent multi-frame methods, as highlighted in Fig. 9. Competing approaches often exhibit artifacts such as blurry edges or incorrect motion in challenging occlusion areas, whereas our autoregressive initialization and multi-stride refinement enable more accurate predictions.

**Influence of Different Temporal Lengths.** We prove the effectiveness of larger temporal receptive fields by comparing estimation differences on occlusions in Fig. 10. Shortening the temporal modeling length ($T = 4$) leads to much worse estimation accuracy on the occluded signal.

**More Visualizations on Spring.** We supplement another visualization sample in Fig. 11. From the figure, our estimated flows in the dynamic eyes are more accurate compared to prior methods like MemFlow and StreamFlow.

Table 7: **Sub-sequence Benchmark results on Spring.** The 1px metrics for 11 categories are listed. Best results are respectively highlighted as first, second.

| | Method | 1px in Each Category | | | | | | | | | | |
|---|---|---|---|---|---|---|---|---|---|---|---|---|
| | | low-det. | high-det. | matched | unmat. | rigid | non-rig. | not sky | sky | s0-10 | s10-40 | s40+ |
| NO FINE-TUNE | PWC-Net (Sun et al., 2018) | 82.268 | 81.747 | 82.069 | 90.400 | 82.817 | 78.090 | 81.575 | 92.761 | 81.402 | 82.189 | 89.693 |
| | FlowNet2 (Ilg et al., 2017) | 6.346 | 64.061 | 5.691 | 48.892 | 3.711 | 29.404 | 6.039 | 16.908 | 1.862 | 5.816 | 49.693 |
| | RAFT (Teed & Deng, 2020) | 6.426 | 64.087 | 5.999 | 39.481 | 4.107 | 27.088 | 5.250 | 30.183 | 3.134 | 5.301 | 41.403 |
| | GMA (Jiang et al., 2021) | 6.699 | 66.203 | 6.281 | 39.892 | 4.276 | 28.247 | 5.614 | 29.263 | 3.645 | 5.389 | 40.327 |
| | GMFlow (Xu et al., 2022) | 9.935 | 76.613 | 9.060 | 63.949 | 6.800 | 37.258 | 8.952 | 31.680 | 5.412 | 9.901 | 52.944 |
| | FlowFormer (Huang et al., 2022) | 6.144 | 64.219 | 5.766 | 37.294 | 3.527 | 29.084 | 5.500 | 21.858 | 3.381 | 5.530 | 35.344 |
| | MemFlow (Dong & Fu, 2024) | 5.394 | 63.348 | 5.107 | 32.755 | 3.293 | 24.422 | 4.494 | 24.990 | 2.918 | 4.820 | 32.071 |
| | StreamFlow (Sun et al., 2024) | 4.869 | 59.550 | 4.559 | 32.343 | 2.865 | 22.987 | 4.435 | 17.059 | 2.597 | 4.492 | 29.067 |
| | MEMFOF (Bargatin et al., 2025) | 3.254 | 58.072 | 3.049 | 26.384 | **1.510** | 19.416 | 3.708 | **1.961** | 1.315 | 4.574 | 20.081 |
| | **ARFlow (Ours)** | **3.180** | **57.251** | **2.973** | **26.375** | 1.675 | **17.526** | **3.395** | 5.505 | **1.084** | **4.244** | **22.009** |
| FINE-TUNE | CrocoFlow (Weinzaepfel et al., 2023) | 4.209 | 60.594 | 3.848 | 34.200 | 2.194 | 22.501 | 4.479 | 5.868 | 1.225 | 4.332 | 33.134 |
| | SEA-RAFT (S) (Wang et al., 2024b) | 3.536 | 61.951 | 3.172 | 34.228 | 1.662 | 20.871 | 3.974 | 2.855 | 1.264 | 4.871 | 23.378 |
| | SEA-RAFT (M) (Wang et al., 2024b) | 3.323 | 60.986 | 3.025 | 31.058 | 1.561 | 19.769 | 3.757 | 2.616 | 1.241 | 4.760 | 21.237 |
| | MemFlow (Dong & Fu, 2024) | 4.119 | 61.703 | 3.742 | 35.115 | 2.391 | 20.306 | 3.934 | 12.809 | 1.305 | 4.437 | 31.184 |
| | StreamFlow (Sun et al., 2024) | 3.790 | 61.297 | 3.424 | 34.304 | 1.986 | 20.544 | 3.986 | 6.678 | 1.236 | 4.381 | 27.935 |
| | DPFlow (Morimitsu et al., 2025) | 3.102 | 56.941 | 2.859 | 27.563 | 1.500 | 18.132 | 3.522 | **2.218** | 1.188 | **3.998** | 20.786 |
| | MEMFOF (Bargatin et al., 2025) | 2.947 | 57.246 | 2.751 | **25.551** | 1.446 | 17.236 | 3.327 | 2.723 | 1.084 | 4.202 | 19.270 |
| | **ARFlow (Ours)** | **2.926** | **56.655** | **2.717** | 25.955 | **1.436** | **17.108** | **3.325** | 2.362 | **1.079** | 4.160 | **19.133** |

Table 8: **End-point-error of flow prediction on FlyingThings3D (Final) Mayer et al. (2016), Sintel (Final), and KITTI-15.**

| Method | FlyingThings3D (Final) | Sintel (Final) | KITTI-15 |
|---|---|---|---|
| Warped Oracle | 14.76 | 5.76 | - |
| MemFlow Dong & Fu (2024) | 15.70 | 6.23 | 12.95 |
| OFNet Ciamarra et al. (2022) | 13.76 | 6.03 | 12.43 |
| MemFlow-P Dong & Fu (2024) | 7.56 | 5.38 | 8.82 |
| Ours | **6.79** | **4.63** | **7.25** |

**Zero-shot Generalization on Waymo and Nuscenes.** Moreover, we visualize zero-shot generalization on Waymo (Sun et al., 2020) and NuScenes (Caesar et al., 2020). Importantly, these datasets are not used during training, yet our ARFlow still achieves high-quality optical flow predictions. As shown in Fig. 12, Fig. 13, Fig. 14, and Fig. 15, our method maintains robust performance in diverse driving scenarios, confirming its strong generalization ability beyond the training distributions.

# E Screenshots for Benchmark results on KITTI, Sintel, and Spring

We include official benchmark screenshots for KITTI-15, Spring, and Sintel in Fig. 16, Fig. 17, and Fig. 18, retrieved on September 23, 2025 (KITTI-15 and Spring) and September 25, 2025 (Sintel).

On *KITTI-15 (Geiger et al., 2013)*, ARFlow ranks **first among all optical flow methods**, achieving the best results on both All and Non-Occ pixels. Note that methods ranked above ours on the leaderboard mainly rely on scene flow or additional 3D information rather than pure optical flow.

On the *Spring* benchmark (Mehl et al., 2023b), ARFlow ranks **first among all open-source methods**, surpassing strong baselines such as MEMFOF and DPFlow.

For *Sintel* (Butler et al., 2012), our method achieves **second place among open-source methods**, matching or outperforming recent state-of-the-art approaches.

These results consistently highlight the advantages of our autoregressive paradigm: scalable temporal modeling, reliable initialization, and robust refinement across datasets.

# F LLM Usage Statement

A large language model (ChatGPT) was used in a strictly limited assistive manner during manuscript preparation. Its usage was confined to: (i) spelling and grammatical error checking; (ii) minor phrasing and wording refinement to improve fluency without altering technical meaning, methodology, analyses, or conclusions; and (iii) occasional condensation of repetitive sentences and suggestions

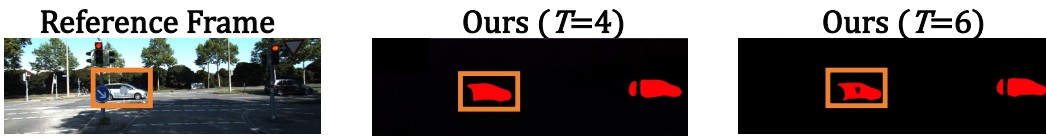

Figure 10: Qualitative results of different temporal lengths on the KITTI benchmark. Sourced from official leaderboard submissions.

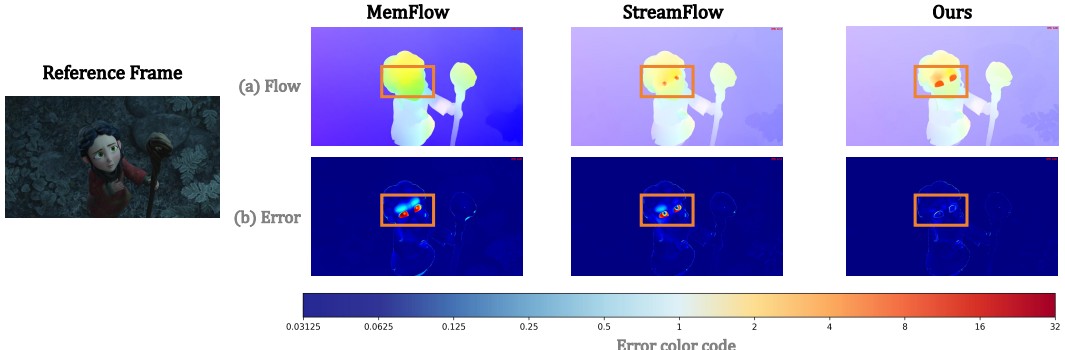

Figure 11: Qualitative results on the Spring benchmark.

for consistent formatting. The LLM did not contribute to research ideation, problem formulation, method design, experimental execution, data processing, result analysis, drafting of technical content, or formulation of conclusions. The LLM is not an author and bears no responsibility for the content.

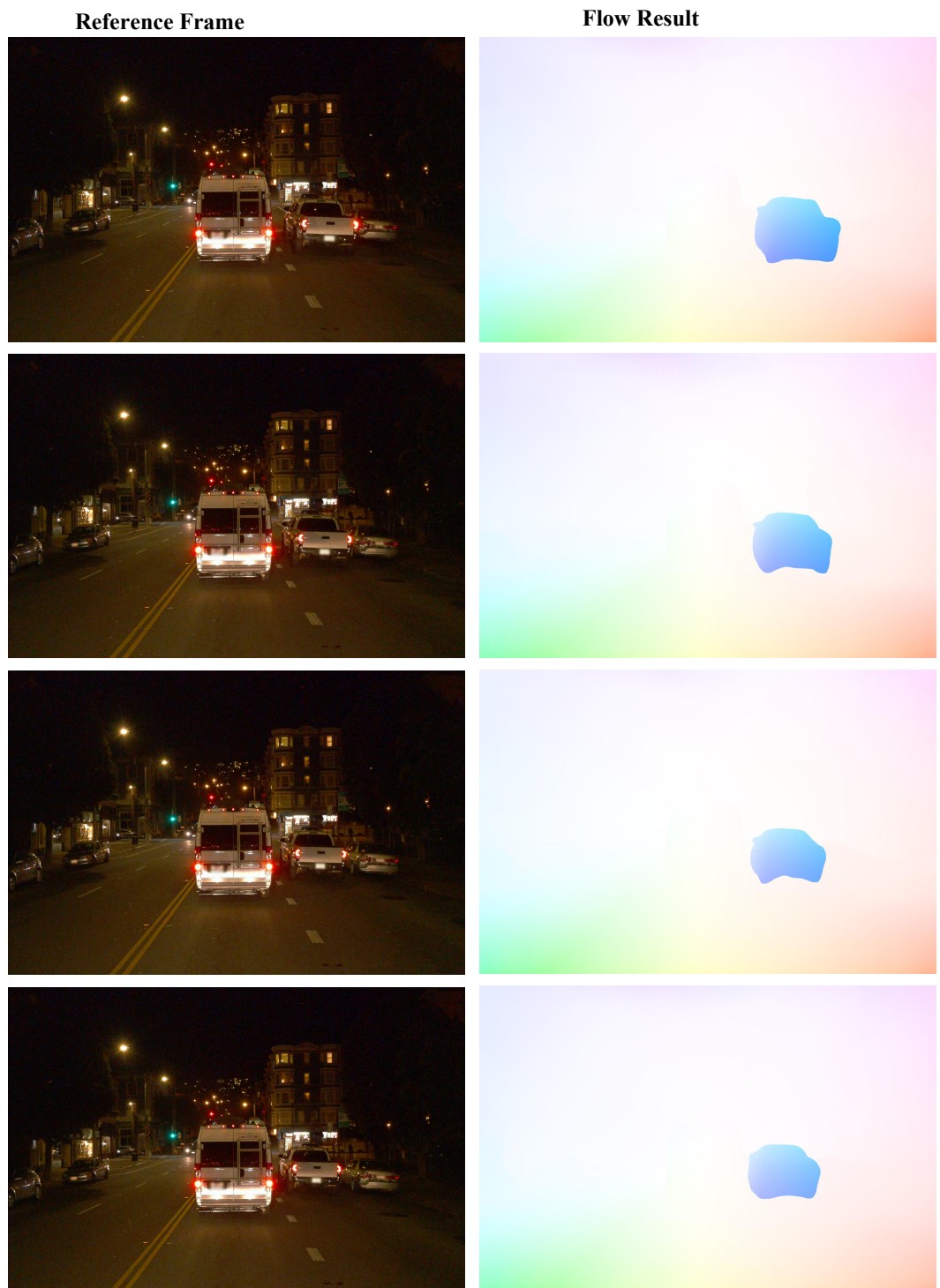

Figure 12: Qualitative results of our method on the Waymo dataset (Sun et al., 2020) (1). Note that our model was not trained on the Waymo dataset.

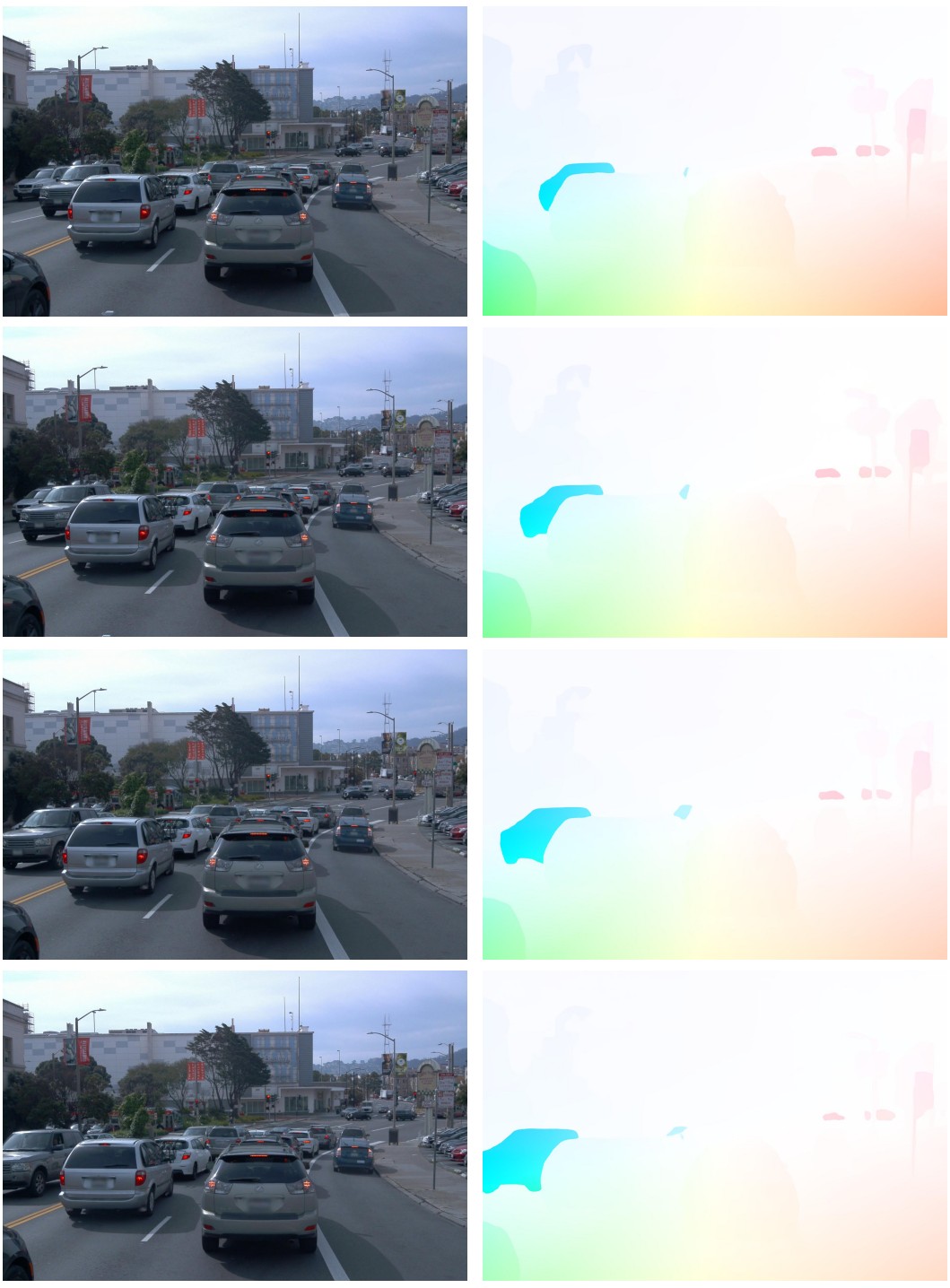

Figure 13: Qualitative results of our method on the Waymo dataset (Sun et al., 2020) (2). Note that our model was not trained on the Waymo dataset.

**Reference Frame**                    **Flow Result**

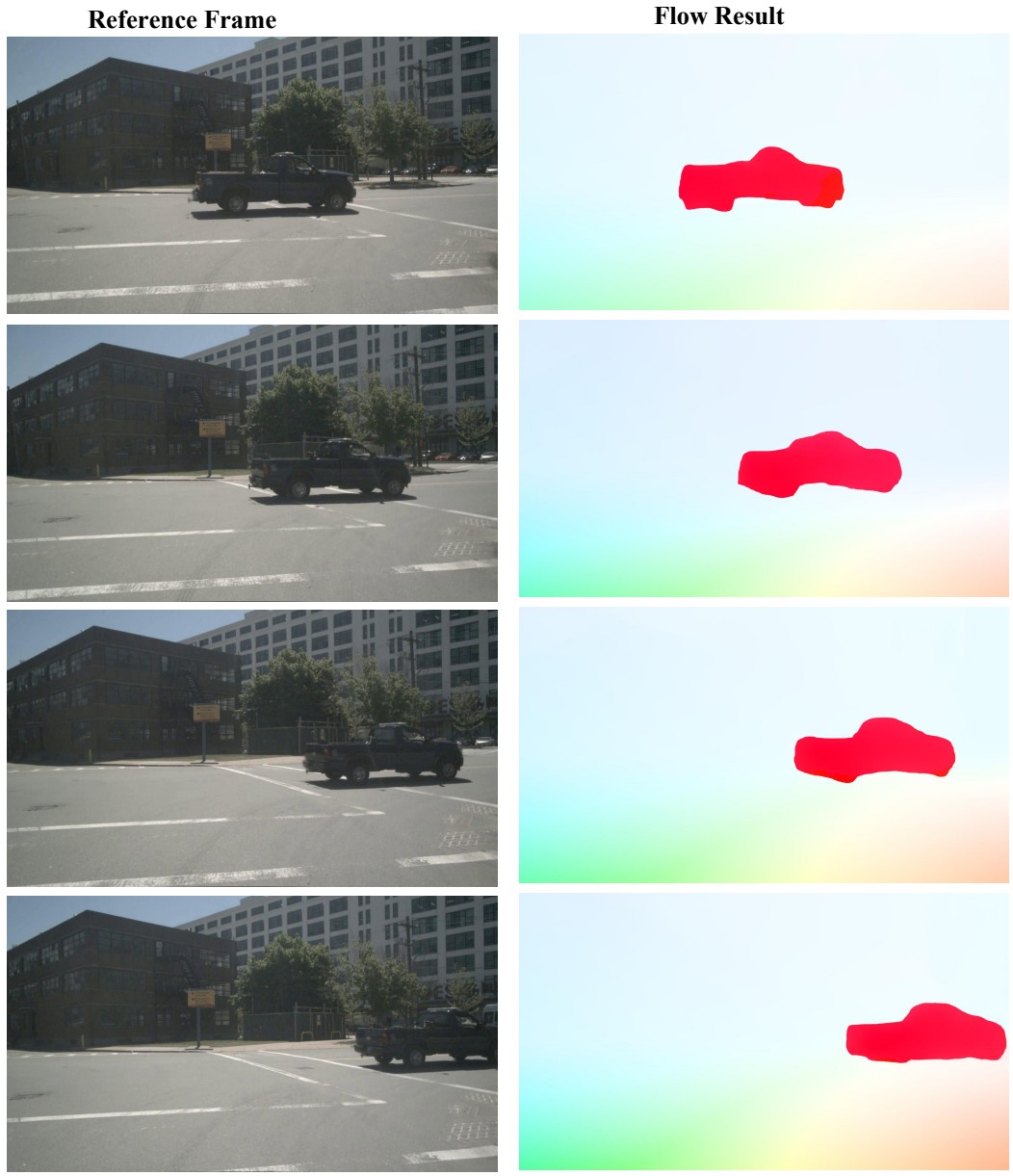

Figure 14: Qualitative results of our method on the NuScenes dataset (Caesar et al., 2020) (1). Note that our model was not trained on the NuScenes dataset.

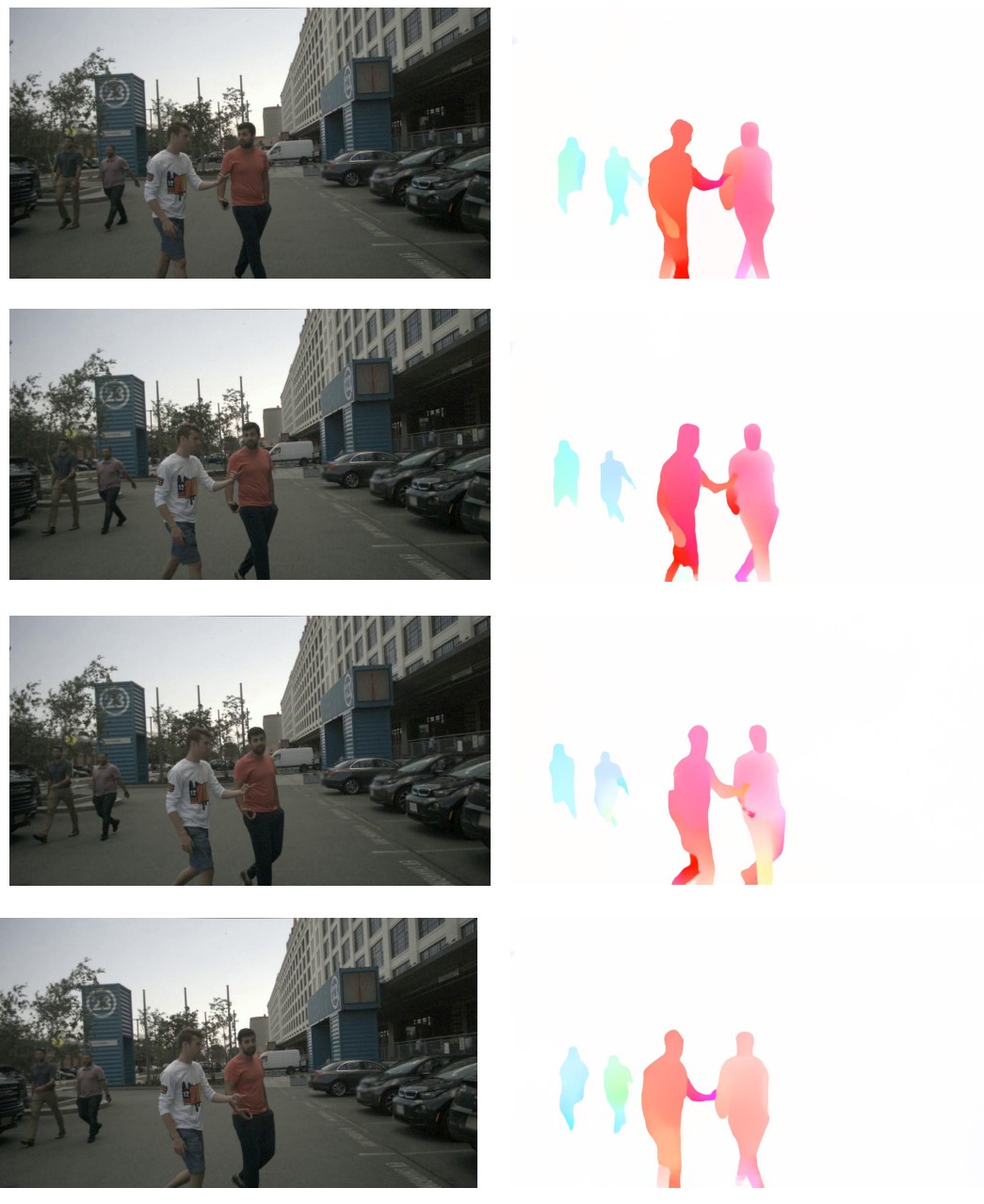

Figure 15: Qualitative results of our method on the NuScenes dataset (Caesar et al., 2020) (2). Note that our model was not trained on the NuScenes dataset.

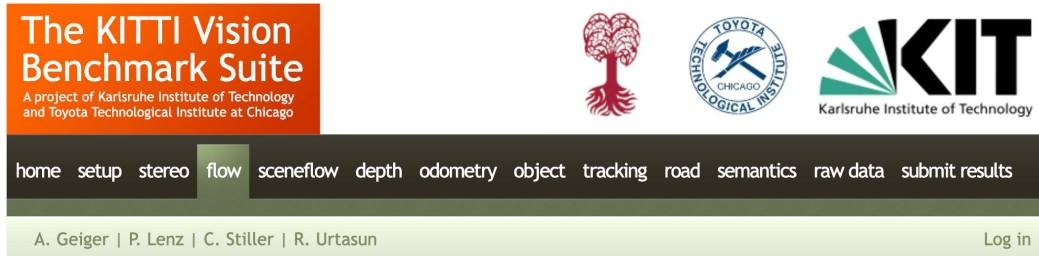

## Optical Flow Evaluation 2015

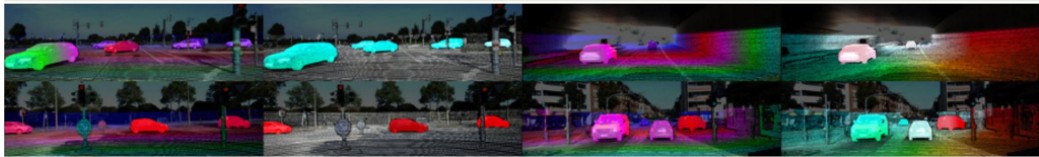

The **stereo 2015** / **flow 2015** / **scene flow 2015** benchmark consists of 200 training scenes and 200 test scenes (4 color images per scene, saved in loss less png format). Compared to the stereo 2012 and flow 2012 benchmarks, it comprises dynamic scenes for which the ground truth has been established in a semi-automatic process. Our evaluation server computes the percentage of bad pixels averaged over all ground truth pixels of all 200 test images. For this benchmark, we consider a pixel to be correctly estimated if the disparity or flow end-point error is **<3px or <5%** (for scene flow this criterion needs to be fulfilled for both disparity maps and the flow map). We require that all methods use the same parameter set for all test pairs. Our development kit provides details about the data format as well as MATLAB / C++ utility functions for reading and writing disparity maps and flow fields. More details can be found in Object Scene Flow for Autonomous Vehicles (CVPR 2015).

Evaluation ground truth [All pixels ▾]        Evaluation area [All pixels ▾]

|  | Method | Setting | Code | Fl-bg | Fl-fg | Fl-all | Density | Runtime | Environment | Compare |
|---|---|---|---|---|---|---|---|---|---|---|
| 1 | SEA-Flow3D + Monster | ⏣ |  | 1.98 % | 5.30 % | 2.53 % | 100.00 % | 0.07 s | GPU @ 2.5 Ghz (Python) | ☐ |
| 2 | MS-RAFT-3D+ | ⏣ | code | 2.22 % | 5.99 % | 2.85 % | 100.00 % | 3 s | GPU @ 2.5 Ghz (Python) | ☐ |
| | J. Schmid, A. Jahedi, N. Senn and A. Bruhn: MS-RAFT-3D: A Multi-Scale Architecture for Recurrent Image-Based Scene Flow. IEEE International Conference on Image Processing (ICIP) 2025. | | | | | | | | | |
| 3 | ARFlow | ⏣ | | 2.48 % | 4.69 % | 2.85 % | 100.00 % | 0.35 s | GPU @ 2.5 Ghz (Python) | ☐ |
| 4 | SplatFlow3D | ⏣ | code | 2.27 % | 6.02 % | 2.89 % | 100.00 % | 0.1 s | 1 core @ 2.5 Ghz (Python) | ☐ |
| | B. Wang, Y. Zhang, J. Li, Y. Yu, Z. Sun, L. Liu and D. Hu: SplatFlow: Learning Multi-frame Optical Flow via Splatting. International Journal of Computer Vision 2024. | | | | | | | | | |
| 5 | SEA-Flow3D+gannet | ⏣ | | 2.08 % | 6.95 % | 2.89 % | 100.00 % | 0.07 s | 1 core @ 2.5 Ghz (Python) | ☐ |
| 6 | OAMaskFlow | ⏣ | | 2.07 % | 7.11 % | 2.91 % | 100.00 % | 0.5 s | 1 core @ 2.5 Ghz (Python) | ☐ |
| 7 | MEMFOF | ⏣ | code | 2.60 % | 4.66 % | 2.94 % | 100.00 % | 0.4 s | GPU @ 2.5 Ghz (Python) | ☐ |
| | V. Bargatin, E. Chistov, A. Yakovenko and D. Vatolin: MEMFOF: High-Resolution Training for Memory-Efficient Multi-Frame Optical Flow Estimation. arXiv preprint arXiv:2506.23151 2025. | | | | | | | | | |
| 8 | CamLiRAFT | ⏣ | code | 2.08 % | 7.37 % | 2.96 % | 100.00 % | 1 s | GPU @ 2.5 Ghz (Python + C/C++) | ☐ |
| | H. Liu, T. Lu, Y. Xu, J. Liu and L. Wang: Learning Optical Flow and Scene Flow with Bidirectional Camera-LiDAR Fusion. TPAMI 2023. | | | | | | | | | |
| 9 | CamLiFlow | ⏣ | code | 2.31 % | 7.04 % | 3.10 % | 100.00 % | 1.2 s | GPU @ 2.5 Ghz (Python + C/C++) | ☐ |
| | H. Liu, T. Lu, Y. Xu, J. Liu, W. Li and L. Chen: CamLiFlow: Bidirectional Camera-LiDAR Fusion for Joint Optical Flow and Scene Flow Estimation. CVPR 2022. | | | | | | | | | |
| 10 | DDVM | | | 2.90 % | 5.05 % | 3.26 % | 100.00 % | | | ☐ |
| | S. Saxena, C. Herrmann, J. Hur, A. Kar, M. Norouzi, D. Sun and D. Fleet: The Surprising Effectiveness of Diffusion Models for Optical Flow and Monocular Depth Estimation. NeurIPS 2023. | | | | | | | | | |
| 11 | WAFTv2-DAv2 | | | 2.98 % | 4.94 % | 3.31 % | 100.00 % | 0.24 s | NVIDIA RTX3090 | ☐ |
| 12 | TDFlow | | | 3.00 % | 5.06 % | 3.34 % | 100.00 % | 0.1 s | GPU @ 2.5 Ghz (Python) | ☐ |
| 13 | DF | | | 3.05 % | 5.23 % | 3.42 % | 100.00 % | 0.1 s | 1 core @ 2.5 Ghz (C/C++) | ☐ |
| 14 | CamLiRAFT-NR | ⏣ | code | 2.76 % | 6.78 % | 3.43 % | 100.00 % | 1 s | GPU @ 2.5 Ghz (Python + C/C++) | ☐ |
| | H. Liu, T. Lu, Y. Xu, J. Liu and L. Wang: Learning Optical Flow and Scene Flow with Bidirectional Camera-LiDAR Fusion. arXiv preprint arXiv:2303.12017 2023. | | | | | | | | | |
| 15 | PAFlow | ⏣ | | 2.75 % | 6.86 % | 3.43 % | 100.00 % | 0.53 s | 1 core @ 2.5 Ghz (C/C++) | ☐ |
| 16 | M-FUSE | ⏣⏣ | code | 2.66 % | 7.47 % | 3.46 % | 100.00 % | 1.3 s | GPU | ☐ |

Figure 16: Screenshots from the KITTI-15 optical flow benchmark on the official website, retrieved on September 23, 2025. **Note:** The methods ranked above ours are based on **scene flow** rather than optical flow.

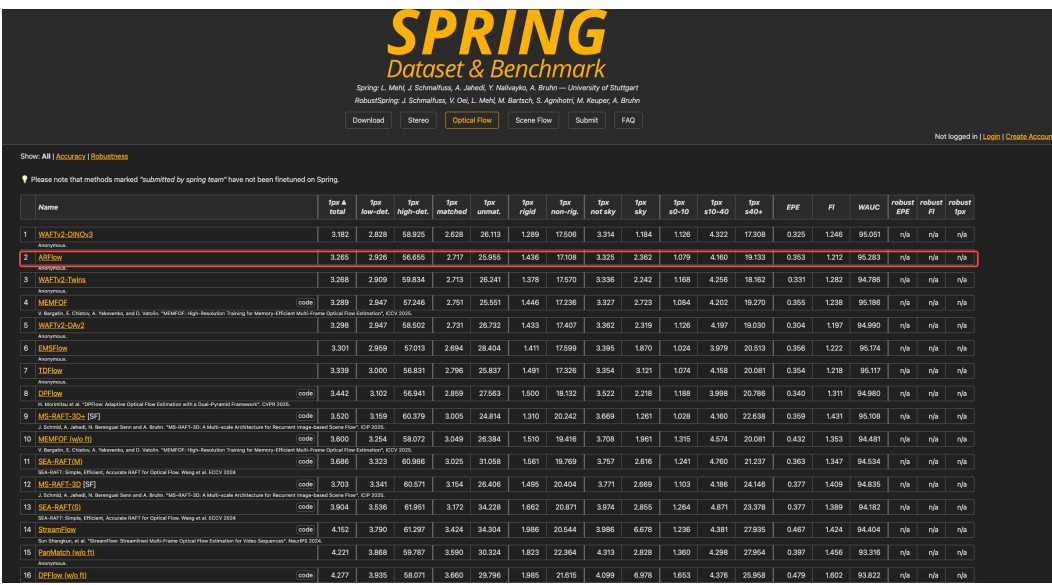

Figure 17: Screenshots from the Spring optical flow benchmark on the official website, retrieved on September 23, 2025. **Note:** To the best of our knowledge (as of September 23, 2025), *WAFTv2* does not have a publicly available paper.

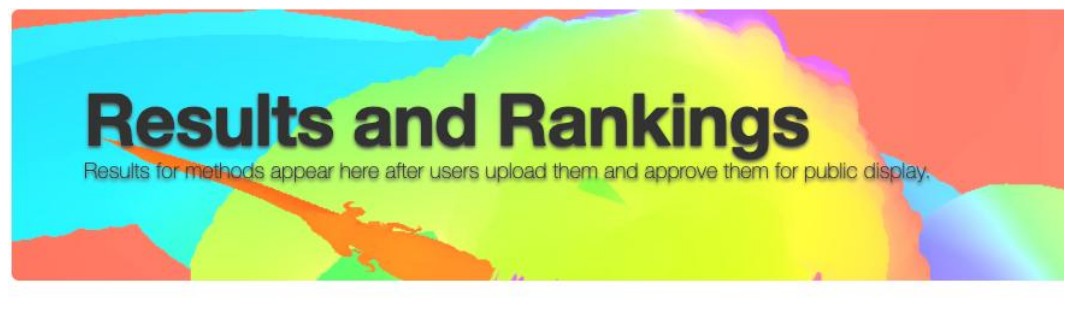

| | EPE all | EPE matched | EPE unmatched | d0-10 | d10-60 | d60-140 | s0-10 | s10-40 | s40+ | |
|---|---|---|---|---|---|---|---|---|---|---|
| GroundTruth [1] | 0.000 | 0.000 | 0.000 | 0.000 | 0.000 | 0.000 | 0.000 | 0.000 | 0.000 | Visualize Results |
| ViCo_VideoFlow_MOF [2] | 1.618 | 0.768 | 8.543 | 1.966 | 0.601 | 0.348 | 0.392 | 1.198 | 8.705 | Visualize Results |
| CFFlow [3] | 1.647 | 0.788 | 8.650 | 2.087 | 0.609 | 0.334 | 0.402 | 1.242 | 8.793 | Visualize Results |
| VideoFlow-MOF [4] | 1.649 | 0.788 | 8.660 | 2.090 | 0.609 | 0.334 | 0.403 | 1.243 | 8.804 | Visualize Results |
| TSA_ [5] | 1.652 | 0.794 | 8.645 | 1.881 | 0.667 | 0.445 | 0.391 | 1.090 | 9.264 | Visualize Results |
| MemoFlow [6] | 1.692 | 0.805 | 8.917 | 2.125 | 0.631 | 0.338 | 0.407 | 1.262 | 9.098 | Visualize Results |
| VideoFlow-BOF [7] | 1.713 | 0.812 | 9.054 | 2.056 | 0.636 | 0.387 | 0.387 | 1.242 | 9.422 | Visualize Results |
| ARFlow [8] | 1.786 | 0.805 | 9.789 | 2.102 | 0.618 | 0.390 | 0.312 | 1.104 | 10.749 | Visualize Results |
| sdex001 [9] | 1.824 | 0.908 | 9.270 | 2.446 | 0.682 | 0.421 | 0.416 | 1.424 | 9.746 | Visualize Results |
| MemFlow-T [10] | 1.840 | 0.874 | 9.710 | 2.233 | 0.671 | 0.370 | 0.467 | 1.351 | 9.828 | Visualize Results |
| StreamFlow [11] | 1.874 | 0.824 | 10.435 | 2.091 | 0.635 | 0.350 | 0.409 | 1.240 | 10.674 | Visualize Results |
| GeoViT [12] | 1.883 | 0.961 | 9.390 | 1.746 | 0.622 | 0.696 | 0.329 | 1.092 | 11.501 | Visualize Results |
| MEMFOF-XL [13] | 1.890 | 0.863 | 10.257 | 2.092 | 0.634 | 0.496 | 0.319 | 1.131 | 11.513 | Visualize Results |
| MEMFOF-L [14] | 1.907 | 0.877 | 10.302 | 2.101 | 0.637 | 0.512 | 0.324 | 1.128 | 11.644 | Visualize Results |
| MemFlow [15] | 1.914 | 0.931 | 9.928 | 2.332 | 0.736 | 0.419 | 0.430 | 1.382 | 10.556 | Visualize Results |
| SemFlow-2view [16] | 1.925 | 0.902 | 10.265 | 2.376 | 0.768 | 0.359 | 0.445 | 1.366 | 10.622 | Visualize Results |
| MEMFOF [17] | 1.942 | 0.890 | 10.513 | 2.121 | 0.641 | 0.524 | 0.332 | 1.125 | 11.900 | Visualize Results |

Figure 18: Screenshots from the Sintel optical flow benchmark on the official website, retrieved on September 25, 2025. **Note:** To the best of our knowledge (as of September 25, 2025), only *VideoFlow* has a publicly available paper; the other higher-ranked methods do not have publicly available publications.

