# OpenReview forum: "ARFlow: Auto-regressive Optical Flow Estimation for Arbitrary-Length Videos via Progressive Next-Frame Forecasting"
_ICLR.cc/2026/Conference — ICLR 2026 Poster_

### Official Review · Reviewer_hJDw · 2025-10-30

**Soundness:** 3
**Presentation:** 3
**Contribution:** 3
**Rating:** 6
**Confidence:** 4

**Summary:**

The paper proposes ARFlow, a novel framework for multi-frame optical flow estimation, which leverages an auto-regressive paradigm to predict optical flow in arbitrary-length videos with marginal computational overhead. ARFlow adopts a progressive next-frame forecasting strategy, with two key components: the Auto-regressive Flow Initialization (AFI) module, which predicts the next-frame initial flow using multiple history flow estimates from the memory bank, and the Auto-regressive Multi-stride Flow Refinement (AMFR) module, which iteratively refines the flow using multi-stride temporal information. The experiment results show that ARFlow achieves state-of-the-art performance on benchmarks like KITTI-2015, Sintel, and Spring and handles long sequences efficiently with consistent memory usage.

**Strengths:**

- The paper proposes using auto-regressive prediction for optical flow estimation, which is novel. By progressively forecasting the next-frame flow based on historical estimates, ARFlow can handle long sequences more efficiently, significantly improving the scalability of optical flow estimation.
- The paper presents good benchmark results, with ARFlow achieving state-of-the-art performance on popular datasets like KITTI-2015, Sintel, and Spring.
- The paper promises to open-source the code after publication, which is a positive development for the open-source community. ARFlow can also serve as a baseline method for subsequent improvements and comparisons.

**Weaknesses:**

- In Line 047, the paper points out that existing multi-frame methods fail to exploit sufficient temporal cues because of Limited temporal receptive fields. However, the temporal intervals used in this paper are similar to those of mainstream methods in both the AFI (T=6) and AMFR (max stride=4) modules. Furthermore, I would question the validity of long-distance information for optical flow estimation tasks. The experiments in this paper also indicate that the benefits diminish over excessively long distances.
- Although the introduction of auto-regressive estimation has reduced the computational overhead of ARFlow, it still falls short of real-time applications.

**Questions:**

- The paper treats optical flow estimation as a streaming problem, which aligns with the inherent properties of the auto-regressive paradigm. However, in most cases, a complete video sequence is already available, so it's natural to anticipate information from later frames. Actually, many existing methods do this. I'm curious why the paper uses unidirectional auto-regression, sacrificing bidirectional information that may further improve accuracy.
- How does ARFlow handle large motions and sudden occlusions? The auto-regressive paradigm relies on historical information, but how does ARFlow handle corner cases when historical information becomes invalid?

---

> ### Author Response · Authors · 2025-11-20
>
> Thank you for your acknowledgment of our work. We supplement the additional descriptions and experiments below.
>
> **Q1. Temporal Receptive Field:** Compared to other mainstream methods, the biggest difference is that our ARFlow **keeps the original temporal sequence during the training process**. However, most of the previous multi-frame methods, such as MemFlow and StreamFlow, interrupt the original temporal sequence by the random training batch shuffle. To be specific, previous methods like MemFlow use the standard batch-wise training strategy, where timestamps among consecutive input batches are not continuous. Thus, **their temporal modeling remains constrained to these segments, like 3 frames within one batch, rather than truly covering arbitrary-length sequences** because of the shuffled timestamps. This train–test inconsistency cannot reasonably be regarded as effective temporal modeling for long-sequence videos. Furthermore, if the temporal range in one batch is increased, they would suffer from infeasible computational burdens as in Figure 2 of the main manuscript.
>
> In contrast, we treat the whole sequential video clip as input and use a clip-wise training method inspired by MOTR (ECCV 2022), rather than the batch-based one. In this case, our memory and supervision can model the temporal information for the whole sequence by sliding along all the frames. Therefore, **our temporal receptive field can be viewed as the whole-sequence awareness for each video**.
>
> **Q2. Validity of Long-Distance Information:** Please allow us to clarify that a relatively longer temporal receptive field is better for the optical flow task. We prove this point by adding a performance comparison on occlusions in Figure 11 of the Appendix. Shortening the temporal modeling length (T=4) leads to much worse estimation accuracy on the occluded signal in the figure, compared to our default settings (T=6). This demonstrates that relatively longer-distance temporal information helps the network recognize occlusions.
>
> We agree that the benefits of leveraging the temporal information will hit the ceiling as the distance increases, meaning that the performance could not be improved infinitely by progressively adding more frames. For example, history frames that are farther than 6-frame intervals would share very rare similarities in terms of motion consistency and semantic geometry. Therefore, our memory length is set to 6, since more frames would not improve accuracy, as in Table 6 of the main paper. The metrics have been saturated when the temporal length exceeds 6. Together with Q1, it is worth mentioning that, as opposed to the other existing multi-frame methods, our method remains flexible and memory efficient in terms of increasing the length of temporal information.
>
> **Q3. Real-time Applications:** Thanks for your insightful views. We acknowledge that there is still a distance according to all existing optical flow algorithms for real-time optical flow applications. As in Table 2, nearly all previous methods cannot achieve real-time (higher than 100 ms) on the Spring benchmark, even for pair-wise optical flow methods.
>
> However, we have demonstrated that our proposed auto-regressive paradigm can serve as a universal paradigm that generalizes well to various baselines as in Table 5 of the revised main manuscript. We believe that if there are more advanced and efficient backbones, our ARFlow can also have real-time application potential.
>
> **Q4. Bi-directional Information:** We have noticed some prior multi-frame methods employ a bi-directional estimation, e.g., VideoFlow (ICCV'23). However, more recent works like MemFlow (CVPR'24) criticized this paradigm. As stated in MemFlow, '**Some multi-frame-based approaches even necessitate unseen future frames for current estimation, compromising real-time safety-critical scenarios.**' Therefore, recent multi-frame optical flow methods mostly adopted a uni-directional manner, which is also followed in our paper. We also add the experiments by comparing the performance of uni-directional and bi-directional settings as:
> | Method                         |Sintel   (train) |                    | KITTI-2015   (train) |          |
> |-----------------|---------------|-----------------|--------------|----------|
> |                    | Clear↓       | Final↓         | EPE↓       | F1-all↓      |
> | **bi-directional**           | **0.88**                 | 2.08            | **2.86**               | 9.39     |
> | **uni-directional (ours)**   | **0.88**                 | **2.07**             | **2.86**               | **9.21**     |
>
> From the table, overall, the performance differences are marginal. We notice that there are slightly worse estimation results measured in F1-all on KITTI-2015 (train) when considering the backward motion. This is partly because the bi-directional time embeddings would confuse the network about the auto-regression direction.

---

> ### Author Response · Authors · 2025-11-20
> **Official Comment by Authors (Part 2)**
>
> **Q5. How ARFlow Handles Large Motions and Occlusions, and How ARFlow Handles Corner Cases when Historical Information Becomes Invalid?** The effectiveness of historical information on addressing corner cases (occlusions, dynamics) has been widely claimed and verified by previous works like MemFlow. Furthermore, our multi-frame settings could dynamically update and retrieve memory information in our designed memory bank, which facilitates the recognition of corner cases by utilizing historical priors. **The historical information embedding in the memory bank will not become invalid on dynamics or occlusions.** For example, objects with large motions introduce inconsistent motions across frames. By integrating multi-frame historical priors, the network may recognize that these dynamic objects possess extremely different motion patterns compared to static or low dynamics. Also, the large motion typically also exhibits a consistent and continuous motion across frames, where **our bank can memorize historical motions to forecast the current one as an accurate initialization**. In contrast, the previous two-frame methods have poor estimation on these corner cases due to the lack of long-range temporal dependencies.
>
> We also demonstrate this claim by qualitative visualizations and quantitative analysis. As in Figures 9 and 10 of the Appendix, our method has more accurate estimations on sudden occlusions and large motions compared to previous multi-frame and two-frame methods. We also analyze the result differences between occluded and non-occluded regions as follows:
>
>
> | Method                    | Clean                   |                      |                     | Final                    |                      |                     | KITTI-15              |              |
> |---------------------------|-------------------------|----------------------|---------------------|--------------------------|----------------------|---------------------|-----------------------|--------------|
> |                           | Unm.↓                  | Mat.↓                | All↓                | Unm.↓                   | Mat.↓               | All↓                | All↓                 | Non-Occ↓     |
> | SplatFlow (IJCV'24)      | 6.06                    | 0.55                 | 1.12                | 10.29                    | 1.06                 | 2.07                | 4.61                  | 2.96         |
> | StreamFlow (NeurIPS'24)  | 6.42                    | **0.38**            | 1.04                | 10.44                    | 0.82                 | 1.87                | 4.24                  | 2.45         |
> | MemFlow (CVPR'24)        | 6.09                    | 0.43                 | 1.05                | 9.93                     | 0.93                 | 1.91                | 4.10                  | 2.56         |
> | MEMFOF (ICCV'25)         | 5.76                    | 0.41                 | 0.99                | 10.51                    | 0.89                 | 1.94                | 2.94                  | 1.97         |
> | DPFlow (CVPR'25)         | 6.36                    | 0.39                 | 1.04                | 10.69                    | 0.91                 | 1.98                | 3.56                  | 2.12         |
> | **ARFlow (Ours)**        | **5.64**                | 0.39           | **0.96**            | **9.79**                 | **0.80**            | **1.78**            | **2.85**             | **1.91**     |
>
>  As shown in the above table, ARFlow improves more on F1-All than on F1-Non-Occ compared to strong multi-frame baselines, indicating that **most of the gains come from occluded regions**. For example, ARFlow reduces F1-All from 4.24$\rightarrow$2.85 for StreamFlow (32.8\% reduction) but only 2.45$\rightarrow$1.91 for Non-Occ (22.0\%). Similar patterns hold for MemFlow (30.5\% vs.\ 25.4\% reductions). A similar trend appears on Sintel: for StreamFlow, ARFlow reduces Clean-All from 1.041$\rightarrow$0.96 (7.8\% reduction) but Clean-Mat only from 0.38$\rightarrow$0.39 (no improvement), and reduces Final-All from 1.87$\rightarrow$1.78 (4.8\%) but Final-Mat only from 0.82$\rightarrow$0.80 (2.4\%).  Here, Unm denotes the error on unmatched (occluded or newly appeared) regions, which contain occlusions and large motions, while Mat corresponds tomatched (non-occluded) regions with more reliable pixel correspondences.
>
> These consistently larger gains on All (which include occluded and large-motion regions) compared to the gains on Mat (mainly non-occluded regions) demonstrate that **ARFlow achieves notably greater improvement in occluded areas**, highlighting its superior capability in handling corner cases like occlusions.

---

> ### Author Response · Authors · 2025-11-27
>
> Dear Reviewer hJDw,
>
> Thank you again for your recognition of our paper's novel framework, improved scalability, and state-of-the-art performance. Also, thanks for your insightful comments and valuable time on our submission. As the discussion period approaches its end, we would like to inquire whether our responses have addressed your concerns.
>
> Based on each of concerns you raised, we have added additional experiments on bi-directional modeling, clarified the temporal receptive field, and incorporated more visualizations on occlusions or dynamics. We hope that this response can effectively address your questions. If any points remain unclear or if you have additional questions, we would greatly appreciate your guidance and will be happy to clarify further.
>
> If you feel that our revisions and explanations have resolved the issues you raised, we would appreciate if you could consider updating your evaluation accordingly. We sincerely look forward to your feedback, and thank you for your time and effort throughout this process.
>
> Yours sincerely,
>
> ICLR Submission 3589 Authors

---

### Official Review · Reviewer_v3yg · 2025-10-31

**Soundness:** 3
**Presentation:** 3
**Contribution:** 4
**Rating:** 4
**Confidence:** 5

**Summary:**

This work proposes a novel multi-frame optical flow estimation network in a so-called "Auto-Regressive" manner, which features flexible and efficient processing of an arbitrary number of input frames and achieves certain accuracy improvements compared to previous methods. This Auto-Regressive approach is not reflected in the internal architecture of the network but rather in the overall pipeline. Nevertheless, it represents a relatively innovative approach in the methodology of multi-frame optical flow estimation.

**Strengths:**

1. This work proposes a novel approach for multi-frame optical flow processing.
2. The method can handle videos of arbitrary length with significantly reduced memory usage and optimized speed.
3. While improving efficiency, the work enhances model performance through effective temporal modeling techniques.

**Weaknesses:**

1. The term "Auto-Regressive" is somewhat nuanced here. Unlike traditional methods like LLMs that perform next-token prediction in discrete spaces, this approach implements autoregression through modifications to the pipeline. Therefore, comparing it with LLMs in related work is less meaningful. It would be more relevant to discuss autoregressive video generation works, as they also incorporate autoregressive changes in their pipelines.

2. Table 1: The font size is too small.

3. Table 2: Should include more Spring metrics. In the Non-finetune section, additional methods like SEA-RAFT could be added.

4. Recommend including more metric evaluations on the Sintel test set.

5. Are there other cases for Figure 6? The hair strand difference in the bottom-left corner is not clearly visible.

**Questions:**

Please refer to the weaknesses. If necessary, I will update the score based on the response and further discussions.

---

> ### Author Response · Authors · 2025-11-19
> **Response to Reviewer v3yg**
>
> Thanks for your acknowledgment of our work's novelty, performance, and efficiency, and thanks for your valuable comments like: 'novel approach', 'significantly reduced memory usage and optimized speed', 'effective temporal modeling techniques', etc. According to your insightful reviews and suggestions, we have revised our paper as follows.
>
> **Q1. Related Work of Autoregressive Video Generation:** Thanks for your suggestion. We have incorporated more recent methods about auto-regressive video generation in Section 2 of the main manuscript, including the latest works from CVPR 2025 and NeurIPS 2025.
>
>
> **Q2. Font Size of Table 1:** We have enlarged the font size of Table 1 of the main manuscript. Thanks for the reviewer's suggestion!
>
> **Q3. More Metrics and Comparison Methods on Spring:** In this revision, we have included more evaluation metrics on the Spring benchmark, including not only the four main metrics: 1px, EPE, Fl, WAUC in Table 2, but also all the 11 sub-class 1px values in Table 3. As in Table 2 and Table 3, our ARFlow surpasses all recent state-of-the-art methods on most metrics, which demonstrates the excellent performance of our proposed method. To the best of our knowledge, these metrics are all the existing metrics on the Spring benchmark, as proved by the official benchmark website (https://spring-benchmark.org/opticalflow).
>
> Furthermore, we follow your suggestions, adding more comparison baselines on the Spring benchmark of the main manuscript, such as GMFlow (CVPR 2022), Win-Win (ICLR 2024), MS-RAFT+ (IJCV 2024), MatchAttention (Arxiv 2025), RAFT3D (CVPR 2021),  M-FUSE (WACV 2023), and MS-RAFT-3D (ICIP 2025). However, the original SEA-RAFT manuscript does not provide its results for the non-finetuning setting in the official Spring benchmark, so we report its metrics on finetuned datasets (SEA-RAFT (S), SEA-RAFT (M)) in Table 2. Thank you for your kind suggestion. These additional metrics and methods further strengthen the quality of our paper.
>
> **Q4. More Metrics on Sintel:** Thanks for your advice! We have incorporated more metrics in Table 1 of the main manuscript, including EPE Matched (Mat.), EPE Unmatched (Unm.), and EPE All (All) for both clean and final test sets on Sintel. As in the table results, our method achieves state-of-the-art performance with the highest accuracy on most metrics. Here, Unm. denotes the error on unmatched (occluded or newly appeared) regions, which contain occlusions and large motions, while Mat. corresponds to matched (non-occluded) regions with more reliable pixel correspondences.
>
> **Q5. Figure 6 and Other Visualization Cases on Spring:** In Figure 6, we wanted to highlight the difference of the boundaries between a person's cheek and hair where our prediction has a sharp boundary. In contrast, previous methods like MemFlow and StreamFlow both estimate a blurry boundary. In this revision, we have narrowed the highlighted region in Figure 6 to avoid misunderstanding. We highly recommend that reviewers refer to the corresponding error visualization in the bottom row for a more obvious comparison.
>
> We also supplement additional visualization samples on Spring for clear comparisons in Figure 12 of the Appendix. From the figure, it is clear that our estimated flows in the dynamic eye regions are more accurate compared to prior methods like MemFlow and StreamFlow.

---

> ### Author Response · Authors · 2025-11-27
>
> Dear Reviewer v3yg,
>
> Thank you again for your recognition of our paper's excellent contribution, high efficiency, and enhanced performance. Also, thanks for your insightful comments and valuable time on our submission. As the discussion period approaches its end, we would like to inquire whether our responses have addressed your concerns.
>
> Based on each of concerns you raised, we have added comprehensive metrics on both Spring and Sintel benchmarks, revised the related work about the auto-regressive video generation, and incorporated more visualizations. We hope that this response can effectively address your questions. If any points remain unclear or if you have additional questions, we would greatly appreciate your guidance and will be happy to clarify further.
>
> If you feel that our revisions and explanations have resolved the issues you raised, we would appreciate if you could consider updating your evaluation accordingly. We sincerely look forward to your feedback, and thank you for your time and effort throughout this process.
>
> Yours sincerely,
>
> ICLR Submission 3589 Authors

---

### Official Review · Reviewer_okv5 · 2025-10-31

**Soundness:** 4
**Presentation:** 3
**Contribution:** 4
**Rating:** 6
**Confidence:** 5

**Summary:**

The paper presents ARFlow, a novel approach to multi-frame optical flow estimation. The proposed method includes two key components: the Auto-regressive Flow Initialization (AFI) and the Auto-regressive Multi-stride Flow Refinement (AMFR), both of which significantly enhance the flow prediction accuracy and computational efficiency. Extensive experiments on various benchmark datasets (KITTI-2015, MPI-Sintel, Spring) demonstrate that ARFlow achieves state-of-the-art performance while maintaining a constant memory usage of 2.1 GB, making it highly efficient.

**Strengths:**

Innovative Approach:The paper introduces a unique solution to multi-frame optical flow estimation by utilizing an auto-regressive model.
State-of-the-Art Performance: ARFlow achieves top-tier results on standard optical flow benchmarks, including KITTI-2015, MPI-Sintel, and Spring.

**Weaknesses:**

1.  The paper does not provide the detailed architecture of its Temporal Transformer. How this transformer process the flow input? Does it employ spatial-temporal attention directly?
2. What’s the inference time per-frame and # model parameter when compared to previous method such as MemFlow?
3. I am also curious about the accuracy of the predicted flow with Temporal Transformer (stride 1). How it performs under the flow prediction setting of MemFlow-P?
4. For the title, I find the phrase of ‘Arbitrary-length videos’ seems to be some wise over-claimed or confused. As previous methods like

**Questions:**

MemFlow typically maintains a memory length of 3 and can also be applied for Arbitrary-length videos. And I cannot tell any other difference with previous methods in terms of videos length from the paper. As for figure 2, I think there may be some minor implementations problems for MemFlow and leads to memory usage increased (, better engineering can solve this problem).
Overall, this is a nice work with solid experiments, I would like to give a rating of accept.

---

> ### Author Response · Authors · 2025-11-19
> **Response to Reviewer okv5**
>
> **Q1. Detailed Architectures of Temporal Transformer:** We have incorporated detailed descriptions of Temporal Transformer in the main paper (Section 4.1). Specifically, we first encode the optical flow and its uncertainty at each time step using 2D convolutions, transforming the input (B, T, 6, H, W) into features (B, T, C, H, W). We then rearrange them as (BHW, T, C), and apply standard multi-head self-attention along the temporal dimension. Moreover, we adopt multi-scale hierarchical Temporal Transformers (downsampled by strides=1, 2, 4 over time) and fuse these sequences with another Temporal Transformer in our AMFR module.
>
> **Q2. Inference Time (on Spring) and Model Parameter:** We compare inference time and model parameters as:
> | Method      | Time / frame (ms) | #Params (M) |
> |-------------|------------------------------|-------------|
> | MemFlow     | 885                          | 6.303       |
> | MEMFOF      | 472                          | 75.783      |
> | StreamFlow  | 929                          | 14.751      |
> | DPFlow      | 990                          | 15.288      |
> | ARFlow-D    | 870                          | 16.172      |
> | ARFlow      | 403                          | 76.497      |
>
> Our method achieves the lowest inference time per frame, demonstrating its high efficiency. For model parameters, our model ARFlow-D (defined as in Table 5 of the main paper) has only slightly larger parameters (0.884M) compared to the baseline method DPFlow, on which we build an auto-regressive paradigm.
>
> **Q3. Flow Prediction Accuracy:** We have evaluated the performance of flow predictions as follows:
> | Method          | FT3D | Sintel | KITTI-2015 |
> |-----------------|------|--------|------------|
> | Warped Oracle   | 14.76| 5.76   | --         |
> | MemFlow         | 15.70| 6.23   | 12.95      |
> | OFNet (ICIAP'22)      | 13.76| 6.03   | 12.43      |
> | MemFlow-P       | 7.56 | 5.38   | 8.82       |
> | **Ours**        | **6.79** | **4.63** | **7.25** |
>
> Compared to previous methods, ARFlow also achieves the best accuracy in the flow prediction task, outperforming the recent method MemFlow-P by 10.2%, 13.9%, and 17.8%, respectively, on three datasets. We attribute this great prediction performance to our auto-regressive paradigm, which naturally captures long-range temporal cues and facilitates the flow extrapolation in a frame-by-frame auto-regression manner.
>
> **Q4. Claim of Arbitrary-length Videos:** Compared to other multi-frame methods, the biggest difference is that our **ARFlow keeps the original temporal sequence during the training process**. However, most of the previous methods, such as MemFlow and StreamFlow, interrupt the original temporal sequence by the random batch shuffle. Prior methods like MemFlow use the standard batch-wise training strategy, where timestamps among consecutively delivered batches are not continuous. Thus, their memory and temporal modeling can only capture a limited range, like 3 frames within one training batch, which fails to extend to arbitrary-length sequences. Overall, even though prior methods can be fed with 'arbitrary-length videos' at inference time, their training still only learns from short segmented clips (e.g., 3-frame batches), so **the temporal modeling remains constrained to these segments rather than truly covering arbitrary-length sequences**; this train–test inconsistency cannot reasonably be regarded as effective temporal modeling of arbitrary-length videos.
>
> In contrast, we treat the whole sequential video clip as input and use a clip-wise training method inspired by MOTR (ECCV'22). Our memory and supervision can model the temporal information for the whole sequence by sliding along all the frames. Therefore, our temporal receptive field can be viewed as **the whole-sequence awareness for arbitrary-length videos**.
>
> Since these details are more engineering-oriented, we avoid detailed descriptions in the main manuscript, but will open-source these details upon publication.
>
> **Q5. Implementations of MemFlow:** We greatly acknowledge the contribution of MemFlow. For implementations of MemFlow, we directly download its open-source code from GitHub and evaluate its GPU memory usage following all its default settings in Figure 2. Furthermore, we agree that there may be better engineering techniques to tackle the increasing memory usage of MemFlow, but it actually doesn't belong to our paper's novelty and workload. In the current MemFlow training pipeline, enlarging the memory length requires feeding more raw frames into the backbone for feature extraction, so increasing the memory length inevitably increases GPU memory usage **during training**, although more careful engineering could reduce this overhead **at inference time**. In contrast, ARFlow keeps the memory length decoupled from the number of frames processed by the heavy backbone, so **increasing the video length almost does not increase GPU memory consumption in either training or inference**.

---

> ### Author Response · Authors · 2025-11-27
>
> Dear Reviewer okv5,
>
> Thank you again for your recognition of our paper's innovative approach, top-tier performance, and contributions. Also, thanks for your insightful comments and valuable time on our submission. As the discussion period approaches its end, we would like to inquire whether our responses have addressed your concerns. Based on each of your concerns, we have added additional experiments, evaluation on the prediction task, and explanations in detail. We hope that this response can effectively address your questions. If any points remain unclear or if you have additional questions, we would greatly appreciate your guidance and will be happy to clarify further.
>
> If you feel that our revisions and explanations have resolved the issues you raised, we would appreciate if you could consider updating your evaluation accordingly. We sincerely look forward to your feedback, and thank you for your time and effort throughout this process.
>
> Yours sincerely,
> ICLR Submission 3589 Authors

---

### Official Review · Reviewer_tPdM · 2025-11-01

**Soundness:** 3
**Presentation:** 3
**Contribution:** 3
**Rating:** 6
**Confidence:** 5

**Summary:**

It presents ARFlow, an autoregressive multi-frame optical flow estimation method designed to address key limitations of existing approaches—limited temporal receptive fields, high computational/memory overhead, and insufficient multi-stride temporal modeling. By introducing an Auto-regressive Flow Initialization (AFI) module and an Auto-regressive Multi-stride Flow Refinement (AMFR) module, it achieves good performance and efficiency.

**Strengths:**

1. Innovative Autoregressive Paradigm: The autoregressive design naturally captures long-range temporal dependencies and supports arbitrary video lengths, which is a good advantage for real-world applications.
2. Efficient Module Design: The AFI module and AMFR effectively modeling both short-term subtle motions and long-term occlusions and achieves good efficiency.
3. Exceptional Performance and Efficiency Balance: According to the paper, ARFlow delivers leading results across three standard benchmarks, outperforming recent strong baselines (e.g., MEMFOF, StreamFlow) while maintaining constant memory usage.

**Weaknesses:**

1. The fonts in Figure 4 and Figure 5 are not uniform.

2. The speed of StreamFlow seems to be inaccurate. If the SIM pipe is not enabled, it will cause many redundant frame calculations and a lot of time waste, please the author check this point again.

3. Considering that this is a brand new structure, it is recommended that the author include more detailed base network structures in the final version and open source them. I am also wondering whether the author has verified that if replacing some pre-trained Transformer methods will bring improvements?

4. More indicators and methods on Spring can be reported.

**Questions:**

Please refer to the weaknesses above.

---

> ### Author Response · Authors · 2025-11-19
> **Response to Reviewer tPdM**
>
> Thank you for your recognition of our paper's novelty, module design, and performance and we have revised our paper below:
>
> **Q1. Figure Fonts:** We have revised the font of Figure 4 and Figure 5 as a uniform version. Please refer to the revised main manuscript.
>
> **Q2. Time of StreamFlow:** Thank you for your meticulous check. In our initial experiments, we followed the official online implementations of StreamFlow but did not enable the SIM module. For the offline case, we have re-run the experiments with SIM enabled and tested different in-batch frame settings with (T=4,5,6,7) as:
> | Method   |  T=4 | T=5 | T=6 | T=7|
> |-------------|-------------|-------------|-------------|-------------|
> | StreamFlow  | 929ms       | 921ms       | 907ms       | 903ms       |
>
> Following the official configuration of StreamFlow, we now report the inference time with SIM (T=4), and we have updated Table 2 in the main manuscript accordingly.
>
>
> **Q3. Detailed Network Structures:** In this revision, we have incorporated detailed descriptions of our network structures in Section 4.1 of the main manuscript. To be specific, our core modules consist of:
>
> (i) **Feature Extraction Backbones**: We use a ResNet--FPN backbone (ResNet-34, dim=512) shared by the context network (cnet) and feature network (fnet). cnet takes concatenated RGB frames to produce 1/16-resolution features and the initial hidden state, while fnet extracts per-frame features for subsequent feature matching.
>
> (ii) **GRU-based Iterative Refinement Backbones**:  On top of these encoders, we adopt a standard RAFT/GMA-style backbone (GMA, ICCV'21) with a 4-stage correlation pyramid (radius=4), a GMAUpdateBlock (GMA, ICCV'21) (num_blocks=2, iters=6 at 1/16 resolution), and a learned convex upsampler for GRU-based Iterative Refinement.
>
> (iii) **Temporal Transformers for Flow Forecasting**: In the Temporal Transformer, we first encode the optical flow and its uncertainty at each time step using 2D convolutions, transforming the input (B, T, 6, H, W) into features (B, T, C, H, W). We then rearrange them as (BHW, T, C), and apply **standard multi-head self-attention along the temporal dimension**. Moreover, we adopt multi-scale hierarchical Temporal Transformers (downsampled by the strides=1, 2, 4 over time) and fuse these sequences with another Temporal Transformer in the Auto-regressive Multi-Stride Flow Refinement module.
>
> In the initially provided code from the supplementary materials, we have released the network modules and structures. We will open source the implementation details and network configurations upon publication.
>
>
>
> **Q4. Replacing by Pre-trained Transformers:** Thanks for your comments. We replace our Temporal Transformer from our original network with two recent pre-trained Transformer backbones: WAN2.2 (5B) [1] and Longcat-video (14B) [2]. The results are as follows:
> | Method                         |Sintel   (train) |                    | KITTI-2015   (train) |          |
> |--------------------------------|----------------|--------------------|---------------------|----------|
> |                                | Clean↓         | Final↓             | EPE↓                | F1-all↓  |
> | WAN2.2 (5B) [1]               | 0.90           | 2.08               | 2.88                | 9.56     |
> | Longcat-video (14B) [2]       | 0.91           | **2.07**           | 2.89                | 9.43     |
> | *w/o pre-trained Transformers (Ours)* | **0.88** | **2.07**           | **2.86**            | **9.21** |
>
> From the table, replacing recent pre-trained transformers cannot bring additional performance gains, while our designed temporal transformer already has effective temporal modeling ability. We have added this ablation study of pre-trained transformers in Table 7 of the Appendix.
>
> [1] https://arxiv.org/abs/2503.20314
> [2] https://arxiv.org/abs/2510.22200
>
>
> **Q5. Listing More Methods and Indicators on Spring:** Thank you for your valuable comments. We have incorporated more comparison methods and evaluation metrics on the Spring dataset, as in Tables 2 and 3 of the main manuscript. The added baseline methods include both optical flow methods, such as GMFlow (CVPR 2022), Win-Win (ICLR 2024), MS-RAFT+ (IJCV 2024), MatchAttention (Arxiv 2025), and also scene flow methods, such as RAFT3D (CVPR 2021),  M-Fuse (WACV 2023), MS-RAFT-3D (ICIP 2025). To the best of our knowledge, we have included nearly all open-sourced existing methods for a comprehensive comparison, as proved by the official Spring benchmark.
>
> In terms of metrics, we also add all the metrics on the Spring official benchmark in the main manuscript, including not only the four main metrics: 1px, EPE, Fl, WAUC (Table 2), but also the detailed 1px values across all 11 sub-classes (Table 3). As in Table 3, our ARFlow surpasses all the recent state-of-the-art methods on most of the sub-classes. To the best of our knowledge, these metrics are all the listed metrics on the Spring official benchmark.

---

> ### Author Response · Authors · 2025-11-27
>
> Dear Reviewer tPdM,
>
> Thank you again for your recognition of our paper's novelty, performance, and innovative paradigm. Also, thanks for your insightful comments and valuable time on our submission. As the discussion period approaches its end, we would like to inquire whether our responses have addressed your concerns. Based on each of your concerns, we have added additional experiments and clarifications in detail. We hope that this response can effectively address your questions. If any points remain unclear or if you have additional questions, we would greatly appreciate your guidance and will be happy to clarify further.
>
> If you feel that our revisions and explanations have resolved the issues you raised, we would appreciate if you could consider updating your evaluation accordingly. We sincerely look forward to your feedback, and thank you for your time and effort throughout this process.
>
> Yours sincerely,
>
> ICLR Submission 3589 Authors

---

### Author Response · Authors · 2025-11-20
**General Response**

We thank all reviewers for their time, constructive feedback, and positive recognition of our work:

**Reviewer tPdM** noted that our method "addresses key limitations" with an "innovative autoregressive paradigm" and "efficient module design";

**Reviewer okv5** regarded our approach as an "innovative approach" and "unique solution by utilizing an auto-regressive model" with "state-of-the-art performance" and "high efficiency";

**Reviewer v3yg** highlighted our "novel approach", "significantly reduced memory usage and optimized speed", and "effective temporal modeling techniques";

**Reviewer hJDw** praised our framework as "a novel framework" that "handles long sequences more efficiently", "significantly improves scalability", and "can serve as a baseline method for subsequent improvements and comparisons".

Furthermore, according to insightful suggestions from each reviewer, we have revised the paper and added new experiments as follows.

**1. For Reviewers tPdM and okv5,** we add a clearer description of the overall network architecture in Section 4.1, including (i) the ResNet–FPN backbone, (ii) the RAFT/GMA-style GRU refinement with a 4-stage correlation pyramid, and (iii) the multi-scale Temporal Transformers in both Auto-regressive Flow Initialization and Auto-regressive Multi-Stride Flow Refinement modules.

**2. For Reviewers tPdM, okv5, and hJDw,** we conduct a more systematic efficiency study: (i) we re-run StreamFlow with SIM enabled under different in-batch settings (T in {4,5,6,7}) and update its inference time in Table 2 of the main paper; (ii) we also add a comparison of inference time and model size on Spring for MemFlow, MEMFOF, StreamFlow, DPFlow, ARFlow-D, and ARFlow, and report the parameter differences between ARFlow-D and DPFlow.

**3. For Reviewers tPdM, okv5, and hJDw,** we add new ablation and analysis experiments: (i) replacing our Temporal Transformer with large pre-trained video Transformers WAN2.2 (5B) and Longcat-video (14B) (Table 7 of the Appendix); (ii) adding optical flow prediction experiments on FlyingThings3D, Sintel, and KITTI-2015 under the MemFlow-P setting; (iii) supplementing ablation studies of uni-/bi-directional autoregression; and (iv) analyzing temporal receptive field by varying the memory length T (Figure 11 of the Appendix).

**4. For Reviewers okv5 and hJDw,** we expand the explanation of our arbitrary-length video setting, contrasting our clip-wise training strategy with the batch-wise training of prior multi-frame methods (MemFlow, StreamFlow).

**5. For Reviewers tPdM and v3yg,** we strengthen the benchmark coverage in terms of both comparison methods and evaluation metrics. (i) On Spring, we add more baselines (GMFlow, Win-Win, MS-RAFT+, MatchAttention, RAFT3D, M-Fuse, MS-RAFT-3D) and report all official metrics, including 1px, EPE, Fl, WAUC, and the 11 sub-class 1px scores (Tables 2–3 of the main paper). (ii) On Sintel, we extend Table 1 with Matched (Mat.), Unmatched (Unm.), and All (All) EPE for both Clean and Final.

**6. For Reviewers v3yg and hJDw,** we refine and extend the qualitative analysis by: (i) adjusting highlighted regions in Figure 6 of the main paper; (ii) adding more visual examples on Spring focusing on challenging regions such as the eye and cheek–hair boundaries (Figure 12 of the Appendix); and (iii) providing additional visualizations and measurements for large motions and occlusions (Figures 9–10 of the Appendix).

**7. For Reviewer v3yg,** we update the related work in Section 2 by adding recent autoregressive video generation methods (including the latest works from CVPR 2025 and NeurIPS 2025).

---

### Author Response · Authors · 2025-12-01
**Summary Written to AC**

Dear Area Chair,

Thank you so much for your valuable time. To facilitate your review process, we have summarized key points regarding the reviewers' comments along with our corresponding responses. I sincerely hope this summary will assist you in your consideration and provide you with a comprehensive understanding of our paper.

1. **Our paper received three acceptance reviews during the initial review period from Reviewers tPdM, okv5, and hJDw.** Although Reviewer v3yg gave a boardline rejection opinion in the initial review stage, he/she chose 'Excellent' in *Contribution*, 'Good' in *Soundness*, and 'Good' in *Presentation* sections, respectively. Reviewer v3yg recognized the novelty of our paper and only suggested some minor revisions like adding some table metrics, adding figures, or revision of the related work, which were all well-resolved in our later response. Furthermore, Reviewer v3yg mentioned he/she had the intention to 'update the score based on the response and further discussions'.


2. **Our paper's novelty** received the high acknowledgment of **all four reviewers**. They all selected 'Excellent' or 'Good' in the *Contribution* part. Furthermore, they also commented like 'Innovative Autoregressive Paradigm' (Reviewer tPdM), 'Innovative Approach' (Reviewer okv5), 'a novel multi-frame optical flow estimation network' (Reviewer v3yg), and 'propose using auto-regressive prediction for optical flow estimation, which is novel' (Reviewer hJDw). Overall, our ARFlow is **the first auto-regressive pipeline for the optical flow task**.

3. **All four reviewers consistently recognized the state-of-the-art performance on three commonly-used benchmarks.** They left comments like 'ARFlow delivers leading results, outperforming recent strong baselines' (Reviewer tPdM), 'ARFlow achieves state-of-the-art performance, highly efficient' (Reviewer okv5), 'significantly reduced memory usage and optimized speed' (Reviewer v3yg), 'ARFlow achieves state-of-the-art performance, and handles long sequences efficiently with consistent memory usage' (Reviewer hJDw).

4. ARFlow also **had other potential advantages or addressed long-standing limitations** that are widely praised by four reviewers. For example, 'address key limitations of existing approaches—limited temporal receptive fields, high computational/memory overhead, and insufficient multi-stride temporal modeling, supports arbitrary video lengths, a good advantage for real-world applications' (Reviewer tPdM), 'significantly enhance the flow prediction accuracy' (Reviewer okv5), 'features flexible and efficient processing of an arbitrary number of input frames' (Reviewer v3yg), 'handle long sequences more efficiently, significantly improving the scalability of optical flow estimation' (Reviewer hJDw).

5. In our authors' response during the discussion stage, we **conducted additional experiments, added more clarifications, incorporated more network details, added the table metrics, and supplemented more figures**. Our response responded all the suggestions from four reviewers in detail and could address their concerns effectively. For more details about our revision, please refer to the *General Response* section and feedback for each reviewer.

Overall, our paper received all four reviewers' recognition in novelty, performance, model design, and scalability. In the discussion period, we also successfully answered all the questions raised by reviewers, which could effectively address their concerns. **We firmly believe that this work can offer a novel insight, have a positive influence, and push the academic development in the optical flow community.**

---

### Meta-Review · Area_Chair_QpwD · 2026-01-07

**Summary:**

## Summary
The paper proposes ARFlow, a novel multi-frame optical flow estimation approach based on an autoregressive transformer models. The proposed approach is novel and experimental results demonstrate state-of-the-art performance on various benchmarks.

## Decision
Most (or all) reviewers appreciate the novelty and the strong performance of the proposed approach, therefore recommend accepting the paper. AC reads all reviews and answers from the rebuttal and agrees with the reviewers' recommendation. AC recommends to accept the paper in its current form.

**Reviewer Concerns:**

Most of the concerns from the reviewers about algorithm, presentation clarity. And in most case, the rebuttal seems sufficiently addressed the questions.

**Reviewer Scores:**

The reviewers scores were 6, 6, 6, 4 originally. There are no further reply / discussion from reviewers after the authors rebuttal.

---

### Decision · Program_Chairs · 2026-01-26

Accept (Poster)